# A shear-dependent NO-cGMP-cGKI cascade in platelets acts as an auto-regulatory brake of thrombosis

Lai Wen [1,5], Susanne Feil[1], Markus Wolters[1], Martin Thunemann [1], Frank Regler[1], Kjestine Schmidt[2], Andreas Friebe[3], Marcus Olbrich[4], Harald Langer[4], Meinrad Gawaz [4], Cor de Wit [2] & Robert Feil[1]

Mechanisms that limit thrombosis are poorly defined. One of the few known endogenous platelet inhibitors is nitric oxide (NO). NO activates NO sensitive guanylyl cyclase (NO-GC) in platelets, resulting in an increase of cyclic guanosine monophosphate (cGMP). Here we show, using cGMP sensor mice to study spatiotemporal dynamics of platelet cGMP, that NO-induced cGMP production in pre-activated platelets is strongly shear-dependent. We delineate a new mode of platelet-inhibitory mechanotransduction via shear-activated NO-GC followed by cGMP synthesis, activation of cGMP-dependent protein kinase I (cGKI), and suppression of $Ca^{2+}$ signaling. Correlative profiling of cGMP dynamics and thrombus formation in vivo indicates that high cGMP concentrations in shear-exposed platelets at the thrombus periphery limit thrombosis, primarily through facilitation of thrombus dissolution. We propose that an increase in shear stress during thrombus growth activates the NO-cGMP-cGKI pathway, which acts as an auto-regulatory brake to prevent vessel occlusion, while preserving wound closure under low shear.

[1] Interfakultäres Institut für Biochemie, University of Tübingen, 72076 Tübingen, Germany. [2] Institut für Physiologie, Universität zu Lübeck, 23562 Lübeck, Germany. [3] Physiologisches Institut, University of Würzburg, 97070 Würzburg, Germany. [4] University Hospital, Department of Cardiology and Cardiovascular Medicine, University of Tübingen, 72076 Tübingen, Germany. [5] Present address: Division of Inflammation Biology, La Jolla Institute for Allergy and Immunology, La Jolla 92037 CA, USA. Correspondence and requests for materials should be addressed to R.F. (email: robert.feil@uni-tuebingen.de)

Platelets are key players in hemostasis and thrombosis after vascular injury. Drugs that inhibit platelet activation and aggregation are commonly used in the management of acute coronary syndromes, but can also cause life-threatening bleeding. An optimal hemostatic response is defined as one sufficient to stop bleeding without unwarranted vascular thrombotic occlusion[1–4]. The importance of local hemodynamic conditions for platelet accumulation at sites of injury is well known and the molecular players that promote platelet activation and thrombus formation under high flow/shear have been studied in great detail[5,6]. In contrast, mechanisms that suppress thrombosis are less well understood. One of the few known endogenous platelet inhibitors is nitric oxide (NO). It is generally accepted that NO is supplied by the endothelium and then binds to and activates NO-sensitive guanylyl cyclase (NO-GC) in platelets, resulting in an increase of intraplatelet cyclic guanosine monophosphate (cGMP)[7,8].

Although genetic studies have demonstrated the importance of NO-cGMP signaling in cardiovascular health and disease in humans[9,10], the underlying cellular and molecular mechanisms are not completely understood. It is long known that platelets express high levels of NO-GC and cGMP-dependent protein kinase I (cGKI), and that the intraplatelet cGMP concentration can be increased in vitro by incubation with NO, or with drugs that stimulate NO-GC or inhibit cGMP degradation by phosphodiesterases (PDEs)[11–13]. However, the (patho-)physiological and therapeutic relevance of platelet cGMP signaling for hemostasis and thrombosis is not clear[14–16]. While it is well established that the NO-cGMP pathway is a major mechanism for platelet inhibition, it is increasingly recognized that this pathway may in fact play a biphasic role in both platelet activation and inhibition. An increase in platelet cGMP has been linked to platelet activation induced by several receptor signaling pathways, including recently discovered platelet pattern-recognition receptor signaling pathways that are triggered by damage-associated molecular pattern molecules such as high mobility group box 1 protein[17].

The uncertainty as to the function of cGMP in platelets is in part related to the fact that it was not possible to monitor dynamic changes of the cGMP concentration in living platelets. Furthermore, many experiments have been performed with isolated platelets in vitro under conditions that do not completely represent the in vivo situation with regard to platelet interactions with immobilized substrates, other cell types, blood flow, and shear forces that affect platelet activity under in vivo conditions[18–20]. In particular, the potential influence of shear stress on the activity of the cGMP cascade in platelets has not been considered until now. To study the role of flow/shear stress for NO-cGMP signaling in platelets, we have generated cGMP sensor mice that enable real-time visualization of cGMP signals during thrombus formation. We find that NO-induced cGMP production in pre-activated platelets is strongly shear-dependent both in a flow chamber assay as well as in two mouse models of arterial injury and thrombosis. The mechanosensitive NO-cGMP-cGKI pathway identified in this study acts as an auto-regulatory brake to prevent vessel occlusion under high shear, while preserving wound closure under low shear.

## Results

### Visualization of cGMP signals in platelets under flow. We first examined the effect of NO on thrombus formation on a collagen-coated surface ex vivo using a flow chamber system. In the presence of the NO-donor DEA/NO (1 μM), murine platelets formed significantly smaller thrombi than untreated controls (Fig. 1a). Inhibition of thrombus formation by DEA/NO was strongly

attenuated when blood from SM-Iβ rescue mice[21] lacking cGKI in platelets was used, demonstrating an anti-thrombotic action of the NO-cGMP-cGKI cascade under flow in vitro. A similar anti-thrombotic effect of DEA/NO was observed in human platelets under flow (Supplementary Fig. 1). The residual weak, but statistically significant ($P = 0.04$, Student's $t$-test), inhibition of cGKI-deficient thrombi by DEA/NO (Fig. 1a) indicated a minor cGMP-cGKI independent anti-thrombotic activity of NO.

To visualize cGMP signals in real time in living platelets under flow and to correlate the spatiotemporal cGMP profile with platelet behavior, we used transgenic cGMP sensor mice[22] that express a fluorescence resonance energy transfer (FRET)- based cGMP sensor, cGi500, in platelets. In these mice, a Cre-activatable loxP-modified cGi500 construct is integrated into the Rosa26 locus and driven by the CAG promoter. In the present study, we used mice with either ubiquitous sensor expression (cGi500-L1 mice) or Pf4-Cre-activated[23] sensor expression specifically in megakaryocytes/platelets (cGi500-L2^fl/fl; Pf4-Cre^tg/+ mice) (Supplementary Fig. 2a). The cGi500 sensor contains a cGMP-binding domain flanked by CFP and YFP. Upon excitation of CFP, part of the emitted energy is transferred via FRET to YFP, which is then also excited. When cGMP binds to the sensor, its FRET efficiency is decreased. Thus, the ratio of CFP over YFP emission (F480/F535) indicates the intracellular cGMP concentration (Supplementary Fig. 2b).

Initially, we have established cGMP imaging in cGi500-expressing platelets adhered to collagen in a flow chamber (Supplementary Fig. 3a). Indeed, strong YFP fluorescence was observed in aggregated thrombi as well as single platelets (Supplementary Fig. 3b and Supplementary Fig. 5a, b). Note that YFP fluorescence indicates the expression level of cGi500, but not the intracellular cGMP concentration. The thrombi in the flow chamber stained positive for the platelet activation marker P-selectin (Supplementary Fig. 3c) confirming that the platelets were indeed activated[24]. To validate the system for cGMP imaging, thrombi were continuously superfused with buffer in the absence and presence of test compounds. While application of CNP and ANP (ligands of guanylyl cyclases not known to be expressed in platelets) did not evoke any FRET/cGMP changes, superfusion with DEA/NO triggered fast, robust, reversible, and concentration-dependent FRET changes reflecting an increase in cGMP in platelet thrombi under flow (Supplementary Fig. 3d, e). Pre-incubation with pharmacological inhibitors of PDE2 (BAY 60-7550), PDE3 (milrinone), or PDE5 (tadalafil) potentiated NO-induced cGMP transients demonstrating that these PDEs degrade cGMP in platelets (Supplementary Fig. 3f–h). These data are in line with previous studies of cGMP signaling in platelets[12,13].

### NO-cGMP signaling in pre-activated platelets is shear-dependent. Surprisingly, flow chamber experiments in which we switched the superfusion off and on revealed that NO-triggered cGMP signals in platelet thrombi are dependent on fluid shear stress. As shown in Fig. 1b and Supplementary Movie 1, application of 100 nM DEA/NO at a shear rate of $500 \, s^{-1}$ led to sustained cGMP elevation, but switching off the fluid flow led to a rapid decrease of the intracellular cGMP concentration to the basal level within ~20 s, although DEA/NO was still present. In the presence of DEA/NO, repeated initiation and cessation of flow (flow on/off) led to repeated cGMP increases and decreases, respectively. Based on a calibration scale of the cGi500 sensor (Fig. 1b, upper right)[25], the flow-induced cGMP elevations were robust, reaching a concentration of ≥3 μM cGMP (Fig. 1b). Note that the apparent plateau of the cGMP signal (ratio R, black trace) was most likely due to sensor saturation above 3 μM cGMP[25]. The possibility that the flow-dependent FRET/cGMP changes

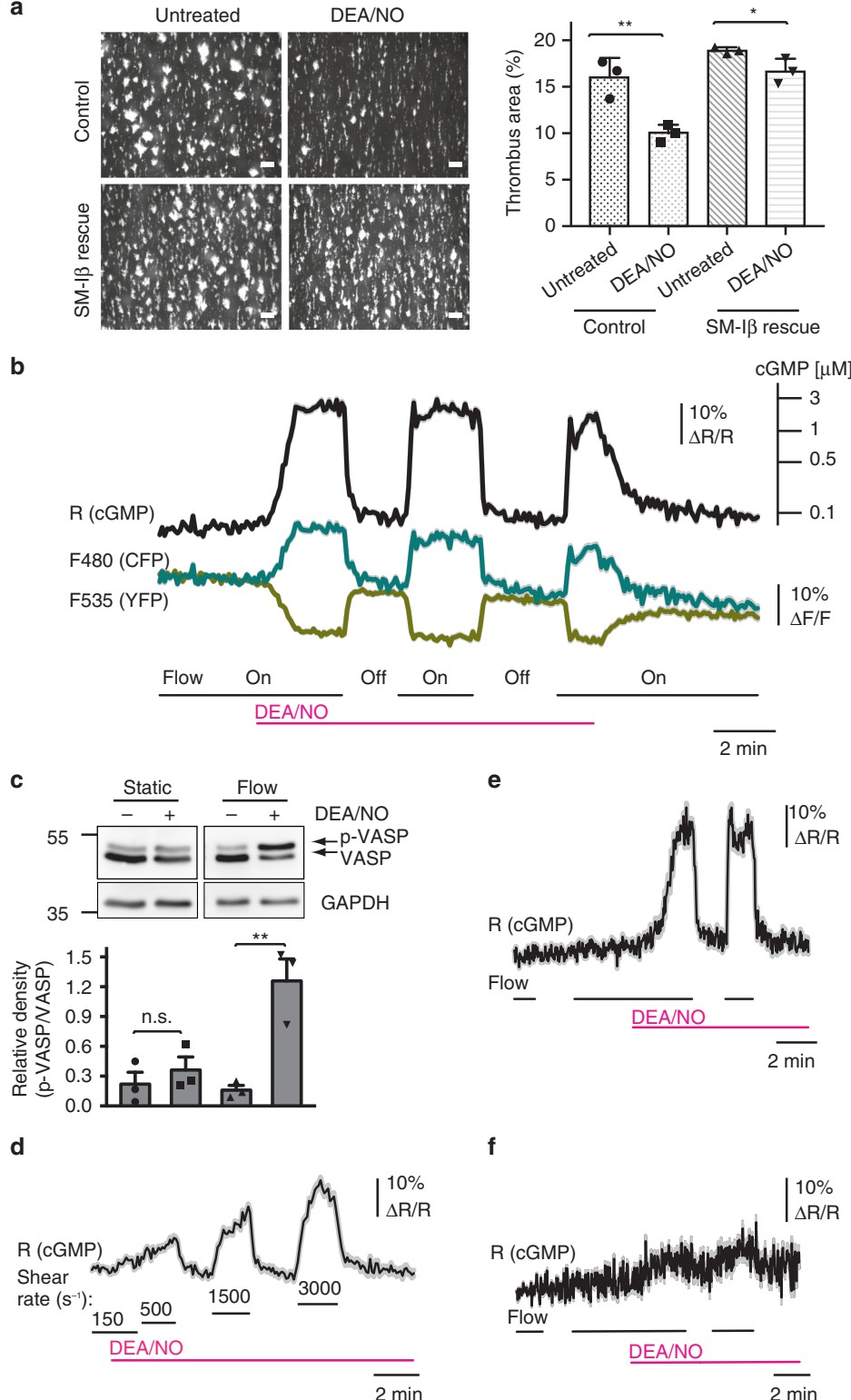

were artifacts was excluded by several control experiments. First, the individual emission intensities for CFP (cyan) and YFP (yellow) changed into opposite directions indicating that a proper FRET change was observed with our ratiometric sensor rather than quenching or fluorescence artifacts (Fig. 1b). Second, the flow-dependent FRET/cGMP signals correlated well with phosphorylation of vasodilator-stimulated phosphoprotein (VASP) (Fig. 1c), a well-known substrate of cGMP-activated

cGKI in platelets[13]. Third, the magnitude of the cGMP signals could be titrated by increasing shear rates (Fig. 1d). Fourth, application of flow alone in the absence of NO did not evoke cGMP signals (Fig. 1e, left). Fifth, flow-induced cGMP signals in the presence of NO were almost absent in cGi500-expressing platelets with a conditional NO-GC knockout (Fig. 1f). The corresponding mice were obtained by crossing platelet-specific cGi500 mice with mice carrying a loxP-flanked allele of

**Fig. 1** A flow/shear-dependent NO-cGMP-cGKI cascade inhibits platelet thrombus formation ex vivo. Thrombi were formed ex vivo by perfusion of mouse blood through a flow chamber over a collagen-coated surface at a shear rate of 500 s$^{-1}$. **a** Thrombus formation by DiOC$_6$-labeled platelets from control mice ($SM22^{+/Iβ}$; $cGKI^{+/L-}$) or SM-Iβ rescue mice lacking cGKI in platelets ($SM22^{+/Iβ}$; $cGKI^{L-/L-}$) in the absence (left) or presence (right) of 1 μM DEA/NO. Data are shown as mean ± SEM (n = 3). Scale bars, 50 μm. **b** FRET/cGMP imaging in cGi500-expressing platelet thrombi superfused with 100 nM DEA/NO at a shear rate of 500 s$^{-1}$. Thrombi were subjected to different shear conditions by switching the buffer flow off and on. Data are shown as mean ± SEM (n = 11 thrombi from one representative experiment out of 10 independent experiments). The cGMP concentration scale (upper right) is derived from in-cell calibration of VSMCs[25]. **c** Western blot detection of VASP phosphorylation (p-VASP) in platelets exposed to buffer without (−) or with (+) 100 nM DEA/NO for 3 min under static or flow (500 s$^{-1}$) conditions. The ratio of p-VASP/VASP was determined in three experiments; GAPDH was used as loading control. **d** FRET/cGMP imaging in cGi500-expressing thrombi superfused with 50 nM DEA/NO at various shear rates (s$^{-1}$). Data are shown as mean ± SEM (n = 8 thrombi from one representative experiment out of three independent experiments). **e**, **f** Platelet thrombi from **e** sensor control mice ($cGi500-L2^{fl/fl}$; $Pf4-Cre^{tg/+}$; $NO-GC$ $β1^{+/fl}$) or **f** sensor mice with platelet-specific NO-GC β1 deficiency ($cGi500-L2^{fl/fl}$; $Pf4-Cre^{tg/+}$; $NO-GC$ $β1^{fl/fl}$) were exposed to flow (500 s$^{-1}$) on/off conditions in the absence or presence of 100 nM DEA/NO. Data are shown as mean ± SEM (n = 10 thrombi). Representative data from one of three independent experiments are shown. The significance level of P values is indicated by asterisks (*P < 0.05; **P < 0.01; n.s., not significant; Student's t-test)

the NO-GC β1 subunit[26] resulting in mice with platelet-specific cGi500 expression and platelet-specific NO-GC deletion ($cGi500-L2^{fl/fl}$; $Pf4-Cre^{tg/+}$; $NO-GC$ $β1^{fl/fl}$). Although the efficiency of Pf4-Cre driven knockouts is very high in platelets[23], we cannot exclude a low level of residual NO-GC expression in some platelets of our conditional NO-GC knockout mice. This might also explain the weak FRET changes triggered by flow and NO in thrombi derived from these mice (Fig. 1f). We observed flow-sensitive cGMP responses also in the presence of higher concentrations of DEA/NO (250 nM, 500 nM) (Supplementary Fig. 4). As expected[27], stimulation with high concentrations of DEA/NO, in particular with 500 nM, induced fast desensitization of the mechanosensitive cGMP response.

Since the results described above were obtained with preformed thrombi consisting of hundreds of aggregated platelets, we tested how individual platelets would respond to shear stress and NO. Indeed, NO-dependent cGMP increases in single adherent platelets were also highly sensitive to flow, both on a collagen- and fibrinogen-coated surface (Supplementary Fig. 5a, b). We further tested the effect of mechanical stress on NO-induced cGMP responses in platelets in suspension. Because our FRET imaging setup is not suitable for cGMP imaging in floating cells, we measured cGMP by a conventional immunoassay. Platelet suspensions were exposed to DEA/NO (50 nM) under static conditions or during mild shaking. Although the shear stress generated under our shaking conditions was not well defined, we detected in these platelets a higher cGMP level than in platelets kept under static conditions (Supplementary Fig. 5c). However, the NO/shear-induced cGMP increase in floating platelets appeared to be weaker than in adherent platelets in a flow chamber. Taken together, our results indicated that shear stress potentiates NO-cGMP signaling in platelets. This mechanosensory system is apparently more efficient in adherent versus floating platelets, but does not require platelet-platelet interactions present in thrombi nor interactions of platelets with specific immobilized substrates.

**NO-GC activity is regulated by fluid shear stress**. Next, we investigated which component(s) of the cGMP pathway is/are regulated by fluid shear stress. The intracellular cGMP concentration in platelets depends on the rates of cGMP synthesis via NO-GC, cGMP degradation via PDEs, and cGMP efflux via cGMP transporters. Thus, the cGMP increase elicited by flow in the presence of NO could be caused by sensitization of NO-GC for NO, inactivation of PDEs and/or inhibition of cGMP efflux. Application of the NO-GC inhibitor ODQ to thrombi in the presence of flow and DEA/NO led to a fast decrease of the intracellular cGMP concentration to basal levels (Fig. 2a,

right). If PDEs were the shear stress-sensitive components, then one would expect that their activity is relatively low under flow and consequently, the fall of cGMP in the presence of NO, ODQ and flow should also be slow. However, we observed a rapid cGMP decrease after acute NO-GC inhibition with ODQ in the presence of flow, with kinetics similar to the decrease of cGMP after flow cessation in the absence of ODQ (Fig. 2a). These data indicated that shear stress increases NO sensitivity of NO-GC rather than decreases cGMP degradation by PDEs. Similar results were obtained with two alternative NO-donors, SPER/NO and DETA/NO, in the presence and absence of flow and ODQ (Fig. 2b, c). Flow-dependent cGMP signals were also observed in the presence of BAY 41-2272 (Fig. 2d) and riociguat (Fig. 2e), so-called NO-GC stimulators that are able to activate NO-GC in the absence of NO[28]. Compared to flow-regulated cGMP transients in the presence of NO, these cGMP signals showed slower off-rates. The relatively slow decline of the cGMP signal after cessation of flow in the presence of BAY 41-2272 might at least in part be due to the fact that this compound also inhibits PDE5[29]. Interestingly, cGMP elevations induced by cinaciguat, a so-called NO-GC activator, were not regulated by flow (Fig. 2f). The differential effects of NO, NO-GC stimulators and NO-GC activators on mechanosensitive cGMP signals are likely related to their different binding sites on NO-GC and modes of action. In contrast to NO and NO-GC stimulators, NO-GC activators activate the enzyme even if it has been oxidized or rendered heme-deficient[7,30]. Inhibition of cGMP efflux with MK-571 did not affect flow-dependent cGMP increases in the presence of NO (Fig. 2g). Together, these data suggested that cGMP synthesis by NO-GC, but not cGMP degradation or efflux, is the major shear-sensitive component of cGMP signaling in platelets.

We also asked whether similar mechanosensitive cGMP pathways might exist in other cell types exposed to physical forces such as vascular smooth muscle cells (VSMCs) that are naturally exposed to wall stress and stretch. Primary VSMCs isolated from mouse aorta express several types of guanylyl cyclases[22]. When we studied these VSMCs in flow chamber experiments similar to those performed with platelets, we also observed shear-sensitive cGMP transients in the presence of NO, but not in the presence of CNP, which activates a different type of guanylyl cyclase, GC-B (Supplementary Fig. 6a). Considering that PDE5 inhibition augments both NO-induced[22] and CNP-induced cGMP pools in VSMCs (Supplementary Fig. 6b), but only the NO-induced cGMP pool was sensitive to flow, it is likely that, similar to platelets, NO-GC but not PDE activity, is the major mechanosensitive component shaping cGMP signals in VSMCs. Indeed, we detected high amounts of the NO-GC β1 subunit as well as cGKI associated with the

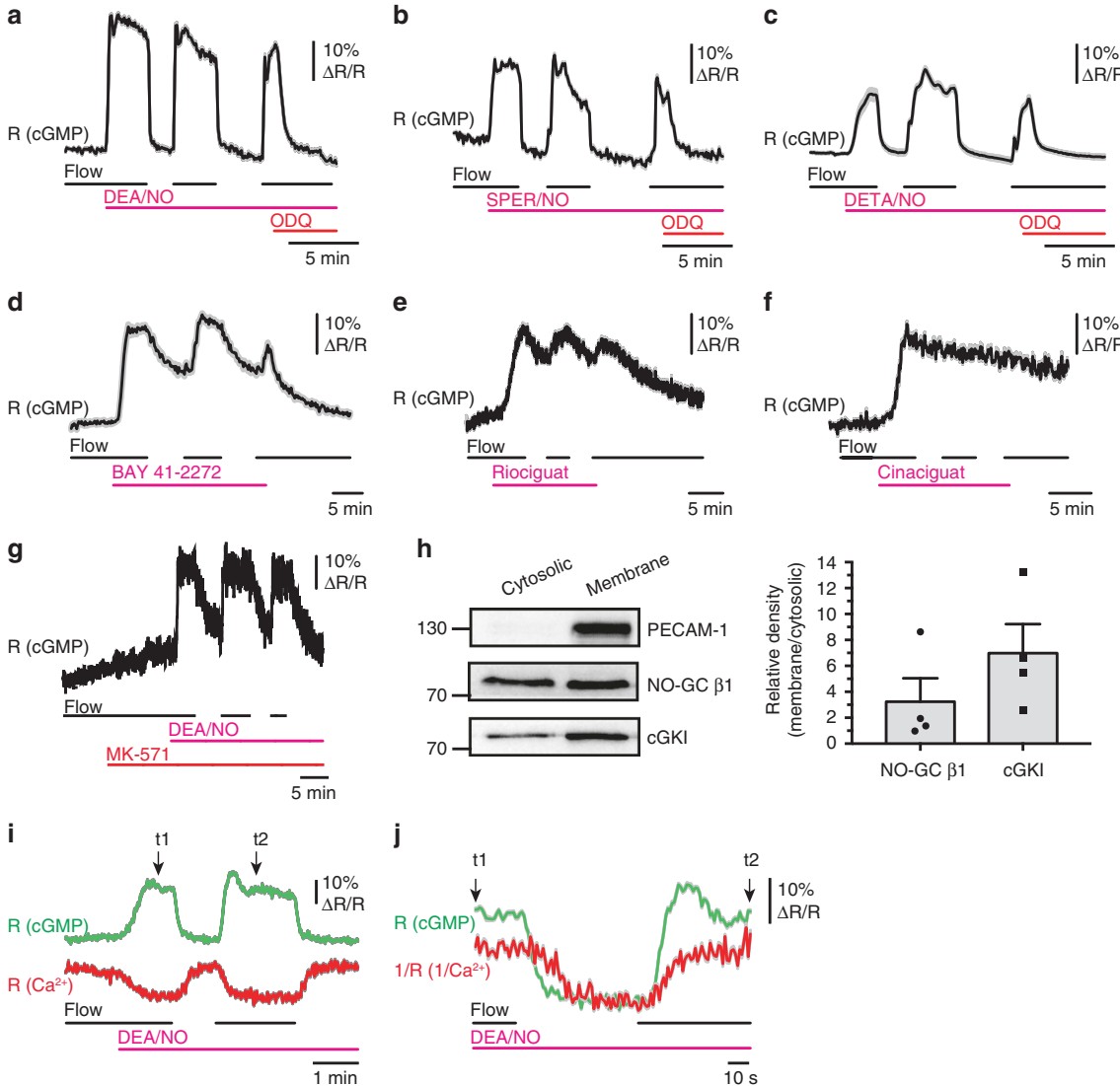

**Fig. 2** NO-GC activity is regulated by fluid shear stress and lowers the intracellular $Ca^{2+}$ concentration. FRET/cGMP imaging of cGi500-expressing platelet thrombi formed on collagen was performed under flow on ($500\,s^{-1}$) and off ($0\,s^{-1}$) conditions. **a–c** Thrombi were exposed to **a** DEA/NO (100 nM), **b** SPER/NO (500 nM), or **c** DETA/NO (10 µM). At the end of each experiment, 20 µM ODQ was applied in the presence of the respective NO donor and continuous flow. **d–f** Thrombi were superfused with the NO-GC stimulators **d** BAY 41-2272 (5 µM) or **e** riociguat (5 µM), or **f** with the NO-GC activator cinaciguat (100 nM). **g** DEA/NO (100 nM)-induced cGMP signals in the presence of the MRP4 inhibitor, MK-571 (1 µM). Data are shown as mean ± SEM ($n = 18$ thrombi in **a**, **b**, **g**; $n = 10$ thrombi in **c**, **d**; $n = 15$ thrombi in **e**, **f**). **h** Western blot analysis of NO-GC β1, cGKI, and PECAM-1 in cytosolic and membrane fractions of mouse platelets. Data are shown as mean ± SEM ($n = 4$). **i** Simultaneous imaging of cGMP (green trace) and $Ca^{2+}$ (red trace) in the presence and absence of flow and 100 nM DEA/NO. These measurements were performed in the presence of 0.2 mM EGTA. Data are shown as mean ± SEM ($n = 25$ thrombi). **j** Traces between time points $t_1$ and $t_2$ of **i** are shown with inverted $Ca^{2+}$ traces to facilitate analysis of the chronological order of changes in cGMP and $Ca^{2+}$. Results are representative of three independent experiments

platelet membrane fraction (Fig. 2h). These findings are consistent with the notion that NO-GC can be associated with the plasma membrane in a state of enhanced NO sensitivity[31]. To test if NO-GC activity is regulated by known mechanotransducers, we performed FRET/cGMP imaging in the presence of Pyr3 (a selective TRPC3 blocker), GsMTx4 (a nonselective blocker of TRPC1, TRPC6, Piezo1, and stretch-activated cation channels), or a $\beta_3$ integrin-blocking antibody. However, none of these compounds affected the flow-regulated cGMP signals in the presence of NO (Supplementary Fig. 7). We suggest that NO-GC, cGKI and other yet unidentified proteins form a signaling complex at the plasma membrane, which converts mechanical force into biochemical signals in platelets, VSMCs, and probably many other cell types.

**Shear-dependent cGMP signals lower intracellular $Ca^{2+}$.** To address the molecular and functional consequences of shear-regulated cGMP signaling, we monitored intrathrombus cGMP and $Ca^{2+}$ concentrations simultaneously in flow chamber experiments. In the presence of NO, initiation of flow induced an increase of cGMP and a decrease of the intracellular $Ca^{2+}$ concentration, while cessation of flow resulted in inverse changes of cGMP and $Ca^{2+}$ (Fig. 2i). cGMP changes preceded $Ca^{2+}$ changes in each case (Fig. 2j). This finding, together with the fact that measurements were performed in EGTA-containing imaging buffer, indicated that mechanosensitive cGMP signals do not require extracellular $Ca^{2+}$ and that activation of shear-dependent cGMP signaling lowers the intracellular $Ca^{2+}$ concentration, presumably by inhibition of intracellular $Ca^{2+}$ release via cGKI

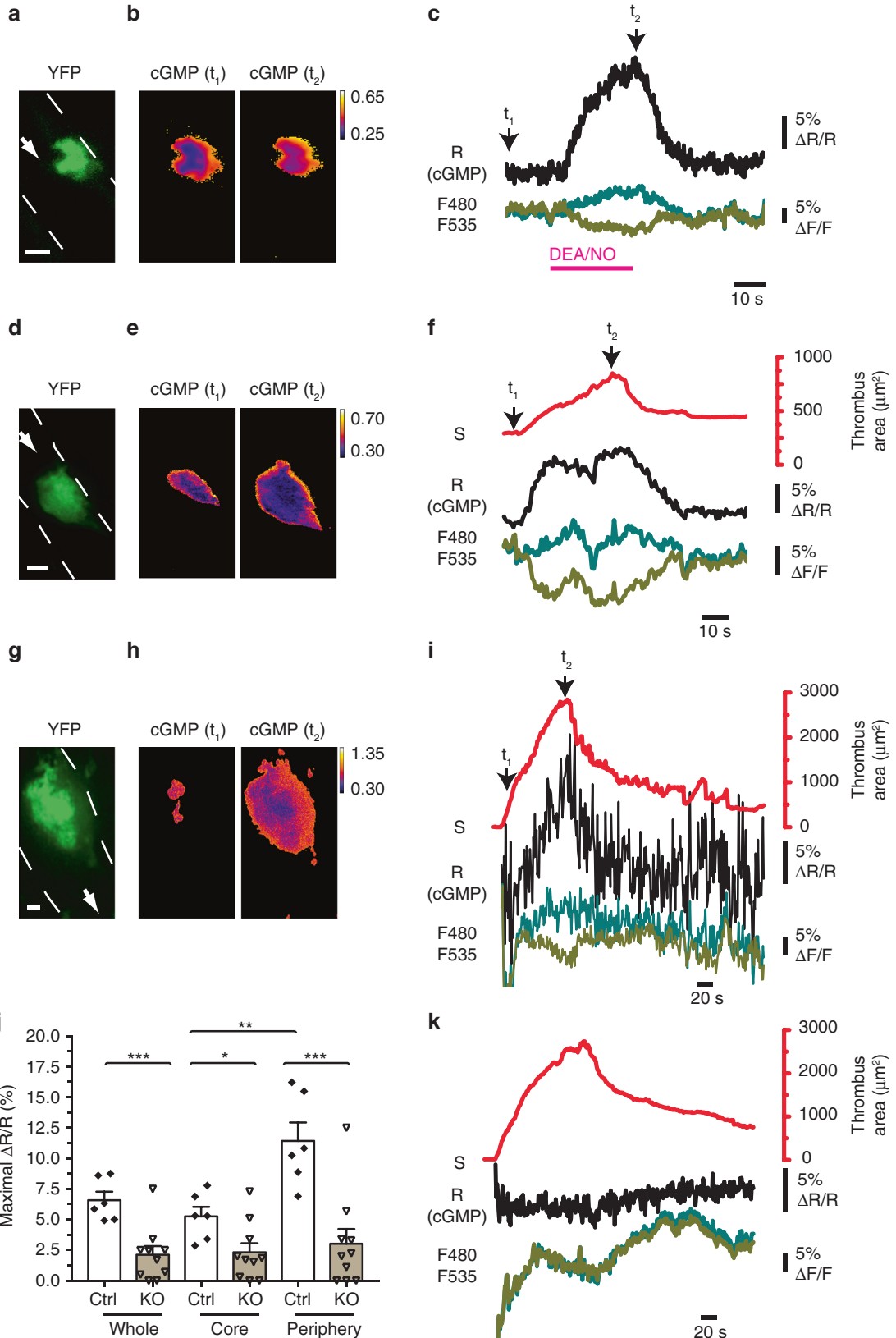

and phosphorylation of IRAG[32]. In line with this mechanism, flow-dependent regulation of $Ca^{2+}$ in the presence of NO was abolished in cGKI-deficient mouse platelets (Supplementary Fig. 8a, b). Moreover, the mechanosensitive NO-cGMP-$Ca^{2+}$ pathway also exists in human platelets (Supplementary Fig. 8c).

Thus, in the presence of NO, fluid shear stress increases intra-platelet cGMP resulting in suppression of $Ca^{2+}$ signals and inhibition of platelet activity, presumably via inhibition of ligand-integrin interaction. The mechanosensitive crosstalk of cGMP and $Ca^{2+}$ signaling likely underlies the anti-thrombotic effect of

**Fig. 3** Thrombus growth is associated with dynamic changes of intraplatelet cGMP in vivo. Platelet cGMP (black traces) and thrombus growth (red traces) were visualized by intravital spinning disk FRET microscopy of cremaster arterioles. Vessel walls are outlined by broken white lines, and white arrows indicate the direction of blood flow. **a–f** cGMP imaging in thrombi formed in platelet-specific sensor mice (*cGi500-L2*fl/fl; *Pf4-Cre*tg/+) after mechanical vessel injury. **a** Sensor fluorescence confirms its specific expression in platelets. **b**, **c** cGMP concentration shown by CFP/YFP emission ratio (R = F480/F535) in a stable thrombus before ($t_1$) and after ($t_2$) application of 10 μM DEA/NO. The cGMP level in the entire thrombus is shown in **c**. The signals at time point $t_1$ and $t_2$ correspond to the images shown in **b**. **d–f** Endogenous cGMP and thrombus area during mechanical injury-induced thrombosis. Note that no exogenous NO was applied. The cGMP level in the periphery of the thrombus is shown in **f**. The signals at time point $t_1$ and $t_2$ correspond to the images shown in **e**. **g–k** Endogenous cGMP and thrombus area during laser-induced thrombus formation. Representative results obtained with sensor control mice (*cGi500-L2*fl/fl; *Pf4-Cre*tg/+; *NO-GC β1*+/fl) (**g–i**) or sensor mice with platelet-specific NO-GC β1 deficiency (*cGi500-L2*fl/fl; *Pf4-Cre*tg/+; *NO-GC β1*fl/fl) (**k**) are shown. Note that no exogenous NO was applied. The signals at time point $t_1$ and $t_2$ (**i**) correspond to the images shown in **h**. **j** Evaluation of maximal FRET signal changes in the entire thrombus as well as in its core and periphery in control (Ctrl) and platelet-specific NO-GC β1 knockout (KO) mice. Data points represent individual thrombi ($n = 6$ and 10 respectively, mean ± SEM, *$P < 0.05$; **$P < 0.01$; ***$P < 0.001$; one-way ANOVA). Color calibration bars in **b**, **e**, **h** indicate relative cGMP concentrations. Scale bars in **a**, **d**, **g** 10 μm

the NO-cGMP-cGKI cascade under flow ex vivo (Fig. 1a; Supplementary Fig. 1).

**Platelet cGMP changes during thrombus growth in vivo.** To demonstrate that the shear stress-dependent NO-cGMP pathway in platelets exists in vivo and inhibits thrombosis in the presence of blood flow, we analyzed cGMP signals and thrombus growth by intravital FRET microscopy in platelet-specific cGMP sensor mice. Thrombus formation is a highly dynamic process with cycles of platelet aggregation and disaggregation, resulting in sequential increases and decreases of thrombus size and shear rates acting on the platelets in the thrombus until the thrombus has stabilized[3,5,19]. We first performed mechanical injury of arterioles of the cremaster muscle with a micropipette and imaged the developing thrombi using an epifluorescence microscope. YFP fluorescence was detected selectively in thrombi indicating platelet-specific expression of the cGi500 sensor in our mouse model (Supplementary Fig. 9a). Superfusion of the cremaster tissue with the NO-releasing drug sodium nitroprusside (SNP) evoked robust FRET/cGMP signals in a concentration-dependent manner (Supplementary Fig. 9b). After this proof-of-principle experiment, we switched to confocal spinning disk FRET microscopy that enables precise mapping of cGMP signals to selected thrombus regions. Thrombus formation was induced by mechanical injury, and after the thrombus stabilized, DEA/NO was applied. Again, administration of an NO donor in the presence of blood flow induced strong increases of the FRET/cGMP signals, which were validated by the fact that individual CFP and YFP traces split into opposite directions (Fig. 3a–c). Importantly, cGMP signals were also observed in growing thrombi in the absence of exogenous NO, and these cGMP signals were stronger in the periphery of the thrombus than in its core region (Fig. 3d–f) (Supplementary Movie 2). The latter finding is consistent with shear-sensitized NO-GC in platelets at the thrombus periphery exposed to flow. In addition, limited diffusion of endogenous NO into the thrombus core may also contribute to the cGMP gradient from the periphery to the core region. Interestingly, the magnitude of the cGMP signals correlated well with the area of the developing thrombus (Fig. 3f). These results indicated that an endogenous NO-cGMP system is active in platelet thrombi exposed to shear stress in vivo.

Our findings are consistent with a model in which thrombus growth leads to increased exposure of aggregated platelets to shear stress[19], which in turn promotes shear-dependent cGMP signaling and inhibition of thrombus growth until it has stabilized. To provide further support for this hypothesis, we used a laser-induced thrombosis model[33], which allows for higher *n*-numbers required for statistical evaluation of data. As in the mechanical injury model, FRET/cGMP imaging of growing thrombi in the absence of exogenous NO showed dynamic changes of the intraplatelet cGMP concentration that paralleled thrombus growth, with the highest cGMP levels observed at the shear-exposed periphery of the thrombus (Fig. 3g–i). Quantitative analysis of FRET/cGMP signals demonstrated a significantly higher cGMP concentration in the thrombus periphery versus its core region in control mice (Fig. 3j). These FRET/cGMP changes were not observed in cGMP sensor mice with a platelet-specific NO-GC deletion (Fig. 3j, k), thus, confirming that the FRET changes recorded by intravital microscopy in growing thrombi of control mice reflected dynamic cGMP signals that depend on the presence of NO-GC in platelets.

**Shear-dependent cGMP facilitates thrombus dissolution.** To study the functional relevance of shear-dependent NO-cGMP signaling for thrombosis in vivo, we compared profiles of thrombus growth over time in control and platelet-specific NO-GC knockout mice. After laser-induced injury of cremaster arterioles, we observed striking differences in thrombus growth profiles in the presence and absence of platelet NO-GC (Fig. 4a, b) (Supplementary Movie 3 and Supplementary Movie 4). While the time to maximal thrombus size and maximal thrombus size was not significantly affected by the genotype, the time required for subsequent thrombus dissolution from peak to 50% thrombus area was significantly increased in the absence of platelet NO-GC (Fig. 4b–e). Overall, deletion of NO-GC in platelets led to increased thrombosis over time (Fig. 4f). In control mice, stabilized thrombi at the end of the experiment were less elongated and significantly smaller than in NO-GC knockout mice (Fig. 4a, b, g). These results indicated that NO-cGMP signaling in platelets in vivo facilitates thrombus stabilization.

**Discussion**
Proper platelet function requires precise temporal and spatial regulation of platelet activation and inhibition. While our study did not specifically examine the role of cGMP during initial phases of platelet activation, we found that NO-induced cGMP generation in already pre-activated platelets is dramatically potentiated by shear stress and results in platelet inhibition at later stages of thrombus formation. Intravital studies performed in the mouse microcirculation showed that thrombi formed after vascular injury develop a characteristic architecture with a core of fully activated, densely packed platelets that is overlaid with a shell of less activated platelets[34,35]. Interestingly, a recent study showed that during the propagation phase, cAMP-dependent protein kinase activity was increased in platelets detaching from the downstream side of the aggregate, suggesting that high activity of cAMP-dependent protein kinase is associated with thrombus resolution[36]. Considering the similarities between

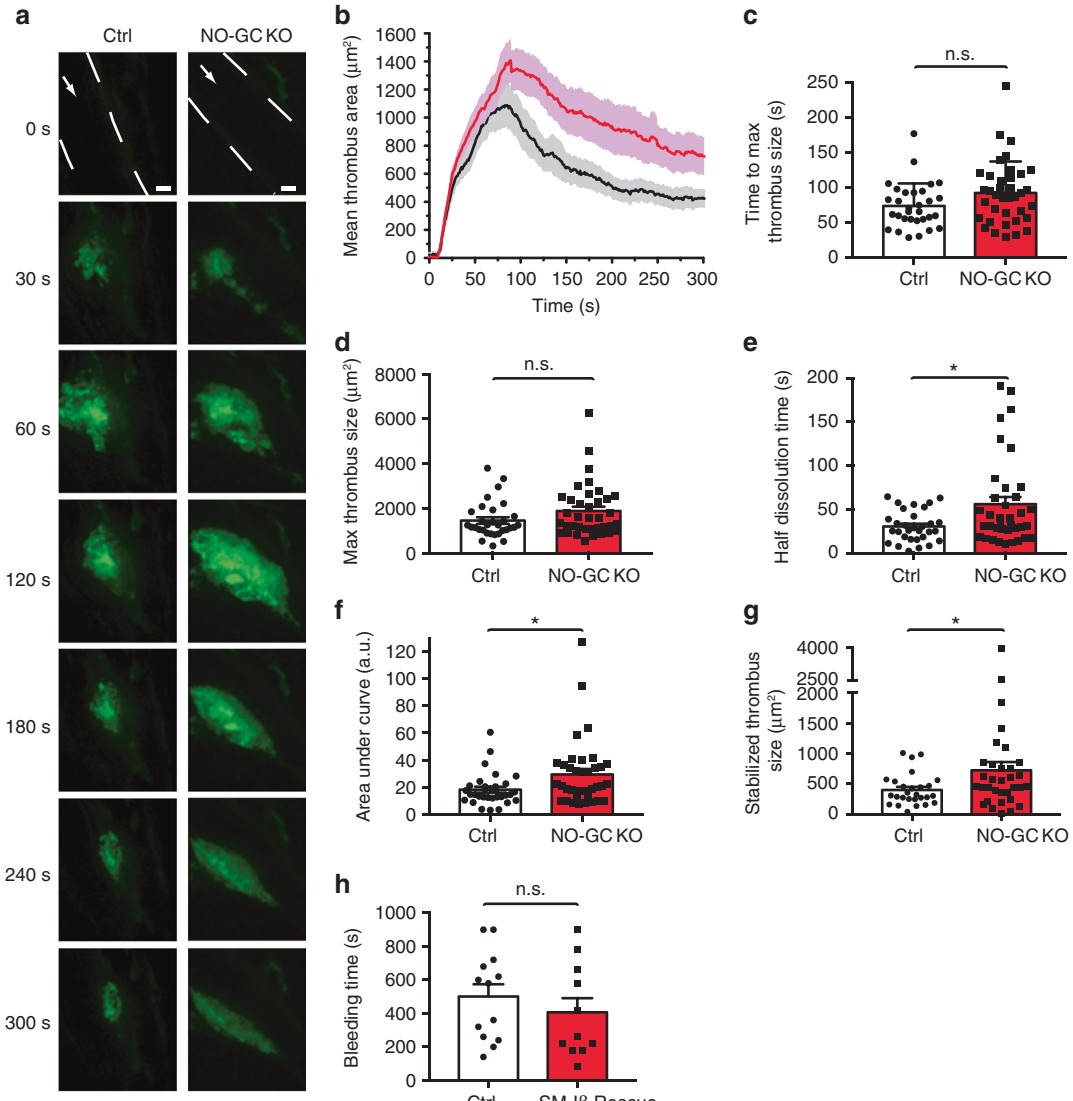

**Fig. 4** Increased thrombosis and thrombus dissolution time in platelet-specific NO-GC knockout mice. **a–g** Laser-induced thrombosis in cremaster arterioles of control mice (Ctrl) (*cGi500-L2*$^{fl/fl}$; *Pf4-Cre*$^{tg/+}$; *NO-GC β1*$^{+/fl}$) and platelet-specific NO-GC β1 knockout mice (NO-GC KO) (*cGi500-L2*$^{fl/fl}$; *Pf4-Cre*$^{tg/+}$; *NO-GC β1*$^{fl/fl}$). **a** Representative fluorescence (YFP) images of platelet thrombi at indicated time points after laser injury. Vessel walls are outlined by broken white lines, and white arrows indicate the direction of blood flow. Scale bars, 10 μm. **b** Profile of thrombus growth shown in **a** depicted as thrombus area over time in Ctrl (black) and NO-GC KO (red) mice. Various aspects of thrombus formation were evaluated: **c** Time to maximum thrombus size; **d** Maximal thrombus size; **e** Thrombus half dissolution time, i.e. the time required after peak maximum to decline to 50%; **f** Area under curve, which reflects integrated thrombus size over time; **g** Size of stabilized thrombi at the end of the experiment (300 s). Data are shown as mean ± SEM; *n* = 31 and 37 thrombi in **b–f** generated in 10 Ctrl and 7 NO-GC KO mice, respectively; *n* = 27 and 33 thrombi in **g** generated in 10 Ctrl and 7 NO-GC KO mice, respectively. **h** Tail bleeding time (mean ± SEM) in control mice (*SM22*$^{+/Iβ}$; *cGKI*$^{+/L−}$) and SM-Iβ rescue mice lacking cGKI in platelets (*SM22*$^{+/Iβ}$; *cGKI*$^{L−/L−}$) (*n* = 13 and 11 mice, respectively). Statistical significance was analyzed in **c–g** using Student's *t*-test (or Welch *t*-test for samples with non-equal variances) and in **h** using Mann–Whitney *U* test. *$P < 0.05$; n.s., not significant

cAMP and cGMP signaling, this finding is consistent with our data indicating that high cGMP concentrations in shear-exposed platelets at the thrombus periphery limit thrombosis primarily through facilitation of thrombus dissolution.

Taken together, our data support a concept of shear-dependent inhibition of thrombosis via the NO-cGMP-cGKI cascade (Supplementary Fig. 10) and also provide a mechanistic explanation for the finding that human gene mutations leading to impaired cGMP generation in platelets are associated with an increased risk of myocardial infarction[10]. Platelets that are recruited to the periphery of a growing thrombus are exposed to increasing shear stress as the thrombus becomes bigger. With increased thrombus size, shear-dependent cGMP synthesis in platelets at the

periphery of a thrombus increases resulting in activation of cGKI and suppression of Ca$^{2+}$-dependent platelet activity. NO that activates shear-sensitized NO-GC in platelets is likely generated by the vascular endothelium, presumably also in a shear stress-dependent manner[37–39]. Thus, NO generation in endothelial cells as well as cGMP synthesis in platelets appears to require shear stress. In this way, the mechanosensitive NO-cGMP cascade provides a negative feedback control of shear stress-induced platelet activation. It acts as an auto-regulatory brake to achieve a balance between activating and inhibiting mechanisms until a thrombus is eventually stabilized. The cGMP brake mechanism is well designed to enable an optimal hemostatic response. It should be most active to limit arterial thrombosis and vascular occlusion

under high shear conditions, which are also known to lead to strong platelet activation, while it should not promote bleeding from injured vessels under low shear. Indeed, in a tail vein bleeding assay, in which platelets are exposed to shear rates ten times lower than in arteries[40], the bleeding time of mutant mice lacking cGKI in platelets was not significantly different from controls (Fig. 4h). Thus, drugs that activate platelet NO-GC in a shear-sensitive manner, such as NO donors or the clinically used NO-GC stimulator riociguat (Fig. 2a–e), could be anti-thrombotics, which, unlike currently used anti-thrombotic therapies, inhibit arterial thrombosis without the life-threatening risk of bleeding. It is tempting to speculate that the mechanosensitive NO-cGMP-cGKI cascade in platelets provides a mechanism for the beneficial effects of NO donors in treating ischemic heart disease.

## Methods

**Animals and reagents.** R26-CAG-cGi500(L1) (cGi500-L1) mice express the cGMP sensor cGi500 globally, and R26-CAG-cGi500(L2) (cGi500-L2) mice express cGi500 after crossing to appropriate Cre mice in a cell type-specific manner[22]. cGMP sensor mice on a mixed 129Sv/C57BL6N genetic background were back-crossed for ten generations to C57BL/6N mice. cGi500-L2 mice were crossed to Pf4-Cre mice[23] to obtain megakaryocyte/platelet-specific cGi500 mice (*cGi500-L2*fl/fl; *Pf4-Cre*tg/+). Mice with platelet-specific cGi500 expression and platelet-specific NO-GC deletion (*cGi500-L2*fl/fl; *Pf4-Cre*tg/+; *NO-GC β1*fl/fl) were generated by crossing conditional NO-GC β1 knockout mice[26] with cGi500-L2 and Pf4-Cre mice. cGKIβ smooth muscle rescue animals (SM-Iβ; genotype: *SM22*+/Iβ; *cGKI*L−/L−)[21], which lack cGKI expression in platelets, were on a C57BL/6N genetic background. Animal experiments were performed in accordance with all relevant ethical regulations and had been approved by the Regierungspräsidium Tübingen (IB 2/15). Atrial natriuretic peptide (ANP), C-type natriuretic peptide (CNP), Pyr3, and fibrinogen was purchased from Sigma Aldrich; fibrillar type I collagen from Nycomed; DEA/NO, SPER/NO, DETA/NO, SNP, and ODQ from Axxora; MK-571, BAY 41-2272, BAY 60-7550, milrinone, tadalafil, and **3,3′**-dihexyloxacarbocyanine iodide (DiOC₆) from Santa Cruz; riociguat from Medchem Express; cinaciguat from Cayman Chemical; GsMTx4 from Alomone labs; anti-β₃ integrin antibody from Biolegend (1:100, 104309) and Fura-2/AM from Calbiochem.

**Thrombus formation in a flow chamber and immunostaining.** Rectangular coverslips (24 × 60 mm) for flow assays were coated with 200 µg/mL fibrillar type I collagen over night at 4 °C and then blocked with 1% BSA solution for at least 30 min at room temperature. 12- to 20-week-old mice were anesthetized without reawakening and 700 µL blood was collected from the retro-orbital plexus into a falcon tube containing 300 µL of heparin in PBS (20 U/mL). Blood was mixed with platelet Tyrode buffer (in mM: 10 HEPES, 137 NaCl, 12 NaHCO₃, 2.7 KCl, 5.5 D-glucose, pH 7.4; supplemented with 0.1% BSA) in a ratio of 2:1. Using a syringe pump (B-Braun), this mixture was perfused through a flow chamber (µ-Slide I⁰·¹ or µ-Slide I⁰·², ibidi) mounted with a collagen-coated coverslip at a shear rate of 500 s⁻¹ (corresponding to a shear stress of 5 dyn/cm²) at room temperature. The shear rate depends on the flow rate and geometry/height of the flow chamber and was calculated according to formulas given by the manufacturer (ibidi). Human blood was drawn from healthy volunteers into tubes containing citrate phosphate dextrose-adenine-1 (Sarstedt) and used for thrombus formation in a flow chamber as described for murine blood. We have complied with all relevant ethical regulations and all human participants gave written informed consent. The study was approved by the institutional ethics committee (Ethik-Kommission an der Medizinischen Fakultät der Eberhard-Karls-Universität und am Universitätsklinikum Tübingen) (474/2014BO2). Before perfusion through the flow chamber, human blood or blood from mice not expressing the cGi500 sensor was fluorescently labeled by pre-incubation with 0.5 µg/mL DiOC₆ for 3 min at room temperature. For immunofluorescence staining, platelet thrombi were fixed for 10 min with 4% formalin, blocked with 5% normal goat serum for 1 h, and then incubated with anti-mouse/rat CD62P (P-selectin) antibody (2 µg/mL; 1:250, Biolegend, 148301) for 1 h. After washing with PBS, cells were incubated with Alexa Fluor 488–conjugated secondary antibody diluted in PBS (1:500, Invitrogen, 51663A) at room temperature for 1 h and embedded in Immu-Mount (Thermo Scientific).

**Primary vascular smooth muscle cells (VSMCs).** Primary VSMCs were isolated from aortae of cGi500-L1 mice[22]. Single cells were obtained by incubation of aortic tissue for 45 min at 37 °C with papain (0.7 mg/mL) followed by 10–15 min with collagenase (1 mg/mL) and hyaluronidase (1 mg/mL), and then plated on rectangular glass coverslips. Cells were grown in culture medium (DMEM with 4.5 g/L glucose, 10% fetal bovine serum, 100 U/mL penicillin and 100 µg/mL streptomycin) at 37 °C and 6% CO₂ for 5–7 days and serum-starved for 24 h before mounting into an ibidi flow chamber for microscopic analysis and FRET/cGMP imaging.

**FRET/cGMP imaging in platelets and VSMCs under flow ex vivo.** FRET/cGMP imaging of cells ex vivo was performed in flow chambers (ibidi) using an epi-fluorescence setup (Supplementary Fig. 3a) based on an inverted Axiovert 200 microscope (Zeiss) equipped with EC Plan NeoFluar 10×/0.3, LD Plan NeoFluar 20×/0.4 air, and Plan NeoFluar 40×/1.3 oil objectives and optional 1.6× Optovar magnification (Zeiss). The imaging setup contains a light source with excitation filter switching device (Oligochrome, TILL Photonics GmbH), a DualView beam splitter with 516 nm dichroic mirror and CFP and YFP emission filters (480/30 nm and 535/40 nm) (Photometrics), and a CCD digital camera (Retiga 2000R, QImaging)[22]. Images were acquired at 0.2 Hz or 1 Hz at room temperature. Adherent cells were exposed to a shear rate of 500 s⁻¹ using a syringe pump (B-Braun). Platelet thrombi and VSMCs were superfused at room temperature with platelet Tyrode buffer and imaging buffer (in mM: 5 HEPES, 140 NaCl, 5 KCl, 1.2 MgCl₂, 2.5 CaCl₂, 5 D-glucose, pH 7.4), respectively. Drugs were applied via two sample loops connected in series and controlled by injection valves (Pharmacia V-7, GE Healthcare).

**Simultaneous FRET/cGMP and Ca²⁺ imaging.** Blood from 12- to 20-week-old cGi500-L1 mice was perfused through a flow chamber (ibidi) for thrombus formation on a collagen-coated surface as described above. Platelet thrombi were loaded with 2.5 µM Fura-2/AM in platelet Tyrode buffer for 45 min at room temperature. A combination of filter and mirror sets was used to allow simultaneous recording of Fura-2/Ca²⁺ and FRET/cGMP signals at room temperature. An Oligochrome light source (TILL Photonics) was used to alternately excite Fura-2 at 340/26 nm and 387/11 nm, and the CFP of cGi500 at 445/20 nm. CFP and YFP emission were recorded with a CCD camera (Retiga2000R, Photometrics). Fluorescence channels were separated using a 470 nm dichroic mirror together with a beam splitter (DualView DV²; Photometrics) equipped with a 516 nm dichroic mirror and CFP and YFP emission filters (480/30 nm and 535/40 nm, respectively). In each acquisition cycle, cells were sequentially excited at 340/26 nm, 387/11 nm and 445/20 nm and emitted light was recorded at 480/30 nm and 535/40 nm. Images were acquired at 0.2 or 1 Hz. For Ca²⁺ imaging of Fura-2 loaded mouse or human platelets not expressing the cGMP sensor, the same setup was used, but without the beam splitter. Samples were sequentially excited at 340/26 nm and 387/11 nm, and Fura-2 emission was recorded using a 410 nm dichroic mirror and 440LP filter[41].

**cGMP ELISA of platelet suspensions.** Murine blood was drawn from the retro-orbital plexus of 12- to 20-week-old mice under anesthesia without reawakening into 7xACD buffer (in mM: 85 sodium citrate, 72.9 citric acid, 110 D-glucose). To obtain platelet-rich plasma, blood was centrifuged at 250×g for 2 min at room temperature. Then, the supernatant was centrifuged at 2000×g for 2 min. Pelleted cells were washed in platelet wash buffer (in mM: 4.3 K₂HPO₄, 4.3 Na₂HPO₄, 24.3 NaH₂PO₄, 113 NaCl, 5.5 D-glucose, pH 6.5) supplemented with 0.1% BSA and resuspended at a density of 6.4 × 10⁸ mL⁻¹ in platelet Tyrode buffer supplemented with 0.1% BSA. Tubes containing suspensions of washed platelets in Tyrode buffer (5 × 10⁷ platelets per sample) were placed at room temperature in racks and incubated with 50 nM DEA/NO for 15 s without vortexing (static condition) or on a vortex shaker (Barnstead Thermolyne Maxi Mix II Vortex Shaker, set to full speed). Control platelets were allowed to rest without DEA/NO and vortex treatment. Then, ice-cold ethanol was added to a final concentration of 66% ethanol. Tubes were put on ice for 15 min and then centrifuged at 18,000×g for 10 min at 4 °C. Supernatants were dried at 90 °C and the cGMP content was determined using an enzyme immunoassay kit (Cyclic GMP EIA Kit, Cayman Chemical) according to the manufacturer's protocol.

**Western blot analysis.** For the stimulation of platelets under static conditions, platelets were prepared from murine blood as described above for cGMP ELISA. Platelet Tyrode buffer or 100 nM DEA/NO was added to isolated platelets (2 × 10⁸ mL⁻¹) in platelet Tyrode buffer in a test tube and incubated for 3 min at room temperature. Then, platelets were spun down at 2000 × g for 1 min, the supernatant was removed, SDS-containing lysis buffer was added, and the lysate heated to 95 °C for 5 min. For the stimulation of platelets under flow, platelet thrombi in a flow chamber were subjected to shear at 500 s⁻¹ in the absence and presence of 100 nM DEA/NO for 3 min at room temperature. Then, platelets were immediately lysed in SDS-containing lysis buffer and heated to 95 °C for 5 min. Lysates were used for detection of VASP phosphorylation by western blotting[41]. Primary antibodies were against VASP/pVASP (rabbit, 1:1000; Cell Signaling, 9A2, 3132), or GAPDH (rabbit, 1:5000; Cell Signaling, 14C10, 2118). The secondary antibody was goat anti-rabbit horseradish peroxidase (HRP)-conjugated IgG (1:5000; Cell Signaling, 7074). For detection of protein localization, cytosolic and membrane fractions were prepared from isolated murine platelets subjected to hypotonic shock[31]. The platelet suspension in platelet Tyrode buffer was centrifuged at 20,000 × g for 1 min at 4 °C and the resulting pellet was lysed in ice-cold hypotonic buffer (in mM: 10 EDTA, 10 NaH₂PO₄, 1 PMSF, pH 7.6). Lysates were then centrifuged at 100,000 × g for 1 h at 4 °C. The supernatant (cytosolic fraction) was collected into a new tube. The pellet (membrane fraction) was washed once with hypotonic buffer, centrifuged at 100,000 × g for 1 h at 4 °C, and then resuspended in hypotonic buffer, so that the final volume of the membrane fraction was equal to the volume

of the cytosolic fraction. After adding SDS-containing lysis buffer, samples were denatured at 95 °C for 10 min. Equal volumes of cytosolic and membrane fractions were analyzed by western blotting using antibodies against PECAM-1 (goat, 1:500; Santa Cruz, sc-1506), cGKI (rabbit, 1:5000)[42] and NO-GC β1 (rabbit, 1:10,000; raised against a protein comprising the 350 N-terminal amino acids of NO-GC β1 fused to glutathione-S-transferase). Uncropped images of all representative blots shown in the manuscript are provided in the supplementary information (Supplementary Fig. 11).

**Thrombosis models and intravital FRET/cGMP imaging in vivo**. The mouse cremaster muscle was prepared for intravital microscopy[43]. Male cGi500-expressing mice were anesthetized by intraperitoneal injection of fentanyl (0.05 mg kg$^{-1}$), midazolam (5 mg kg$^{-1}$), and medetomidin (0.5 mg kg$^{-1}$). Then, the cremaster muscle was exteriorized, cleaned of connective tissue, and spread flat on the pedestal of a custom-built imaging platform. Body temperature was maintained at 37 °C by convective heat throughout the experiment. The cremaster was continuously superfused with saline buffer (in mM: 118.4 NaCl, 20 NaHCO$_3$, 3.8 KCl, 2.5 CaCl$_2$, 1.2 KH$_2$PO$_4$, 1.2 MgSO$_4$) warmed to 35 °C and bubbled with 95%N$_2$/5% CO$_2$. SNP or DEA/NO was diluted 1:100 into the superfusion solution using a roller pump to achieve a final concentration of 3–100 μM or 10 μM on the tissue, respectively. Arterioles with a diameter of 30–60 μm and unperturbed blood flow were selected for study. To induce thrombosis by mechanical injury, the skeletal muscle was carefully removed from the selected arteriole to facilitate subsequent injury of the vessel wall by puncture with a micromanipulator-controlled glass micropipette (1-μm tip). Alternatively, a laser-induced thrombosis model was used. Here, thrombus formation was initiated by injuring an arteriole of the intact cremaster muscle with an ablation laser (teem photonics; wavelength 532 nm; energy per pulse >3 μJ; pulse width <0.5 ns; peak power >6 kW) on a spinning disk microscope. Growing thrombi induced by mechanical injury were imaged with an upright epifluorescence microscope (Axioskop 2 FS, Zeiss) equipped with a water immersion objective (Achroplan 40×/0.75 W), a polychromatic light source (Polychrome V, TILL Photonics) set to 420 nm, and an EM-CCD camera (QuantEM 512, Photometrics), or with a spinning disk microscope. Laser-induced thrombosis was monitored with the spinning disk microscope. The spinning disk setup comprised an upright microscope (Axio Examiner Z1, Zeiss), a Yokogawa CSU-X1 spinning disk confocal scanner, diode lasers of 445 nm, 488 nm and 561 nm, water-immersion objectives (W N-ACHROMAT 10x/0.3, W Plan-APOCHROMAT 20×/1.0 DIC (UV) VIS-IR, W Plan-APOCHROMAT 40×/1.0 DIC VIS-IR; all from Zeiss), an oil-immersion objective (Plan-APOCHROMAT 63×/1.4 Oil DIC; Zeiss) and an air objective (EC Plan-NEOFLUAR 2.5×/0.085; Zeiss), a DualView DV$^2$ beam splitter (Photometrics; equipped with 470/24 nm CFP emission filter, 505 nm dichroic mirror, and 535/30 nm YFP emission filter), an EM-CCD camera (QuantEM 512, Photometrics), and a CCD camera (Spot Pursuit, Diagnostic Instruments). A pE-2 LED system (CoolLED) was used for epifluorescence illumination at 400 nm, 450 nm, 500 nm, and 561 nm. Simultaneous recording of CFP and YFP emission was performed using the EM-CCD camera in combination with the beam splitter. The system was controlled by VisiView software (Visitron Systems). Images were acquired at a rate of 1 Hz with 300 ms exposure.

**Evaluation of imaging data and thrombus growth**. Offline image analysis was done using arivis Browser 2D software (Arivis) or Fiji[44]. For further data evaluation, Microsoft Excel (Microsoft), Graphpad Prism 7 and Origin Pro 2016 (OriginLab) were used. For epifluorescence-based FRET/cGMP imaging, the mean intensity from selected regions of interest of the CFP and YFP channel images was corrected for background fluorescence and is referred to as F480 and F535, respectively. The cGMP-representing fluorescence ratio R was calculated as F480/F535. ΔF480/F480, ΔF535/F535, and ΔR/R traces were generated by normalization to the respective baseline. Since the emission spectrum of Fura-2 overlaps with emission of CFP and YFP, emission of Fura-2 can be collected with the same filters used for the CFP/YFP FRET pair. The images from the 535 nm channel excited at 340 nm and 380 nm were used to analyze Fura-2 emission, processed analogously to F480 and F535 and are referred to as F340 and F380, respectively. To estimate peak areas of ΔR/R traces, the Peak Analyzer Module of Origin was used. Traces were corrected for baseline drifts by subtracting a linear baseline, and peak borders were manually defined. To evaluate FRET/cGMP signals in a dynamic thrombus recorded during intravital imaging, masks were applied to select different regions of interest, i.e. the whole thrombus, its core, or its periphery. Image segmentation was applied based on thrombus brightness using a similar approach as described for the analysis of FRET/cGMP signals in the vessel wall[43]. Measurements with a signal-to-noise ratio below 2.5 were excluded from analysis. The sum of CFP and YFP fluorescence intensities was used to define a dynamic binary mask. Gaussian blur filtering (sigma radius = 2 pixels) was applied to the sum image of CFP and YFP when generating the binary mask. An intensity threshold was manually set to segment the whole thrombus from the background. To further segment the core and periphery of the thrombus, the original thresholded mask for the whole thrombus was divided into two portions, i.e. the mask was further eroded by a user-defined distance (defined as ~20% higher than original threshold intensity). The portion remaining after erosion was taken as thrombus core. The portion removed by erosion was taken as the thrombus periphery. The dynamic masks were

multiplied with the original CFP and YFP images to generate new CFP and YFP time lapse images, respectively. A region of interest was selected to cover the whole thrombus during the time lapse recording. Background was selected in a region without thrombus in the original CFP or YFP image without mask applied. Background corrected F480 and F535 signals were used for FRET evaluation. The profile of thrombus growth was analyzed by measuring thrombus area as a function of time. The area under the curve reflecting thrombosis over time was determined by analyzing peak area of the thrombus growth profile using the Peak Analyzer Module of Origin. In addition, thrombus formation was characterized by the following parameters: time to maximal thrombus size, the magnitude of maximal thrombus size, thrombus half dissolution time, defined as the time from maximal thrombus size to 50% of maximal thrombus size, and size of stabilized thrombus at the end of the experiment.

**Tail bleeding assay**. Litter-matched control and mutant mice were anesthetized without reawakening by intraperitoneal injection of fentanyl (0.05 mg kg$^{-1}$), midazolam (5 mg kg$^{-1}$) and medetomidin (0.5 mg kg$^{-1}$). The tail tip (5 mm) was cut off with a scalpel. Blood drops were collected every 20 s on a filter paper without touching the lesion until blood flow stopped. Bleeding exceeding 15 min was stopped using tissue glue. After completion of the tail bleeding assay, blood was drawn from the retro-orbital plexus of each anesthetized animal and platelet counts were determined using an automated cell counter (Sysmex GmbH). Only mice with a normal range of platelet counts (800–1300 × 10$^6$ mL$^{-1}$) were included for analysis of the bleeding time.

**Statistics**. Data are expressed as mean ± SEM. Statistical differences between more than two groups were analyzed by one-way ANOVA followed by Tukey's multiple comparison test. For parametric data, statistical significance was analyzed using Student's $t$-test (or Welch $t$-test for samples with non-equal variances) following determination of normal distribution and equal variances. For non-parametric data (bleeding time analysis), Mann–Whitney $U$ test was applied. $P$ values < 0.05 were considered significant. The significance level of $P$ values is indicated by asterisks (*$P < 0.05$; **$P < 0.01$; ***$P < 0.001$; ns., not significant). All analyses were performed with Origin software.

## Data availability
The data that support the findings of this study are available from the corresponding author upon reasonable request.

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

## Acknowledgements
We thank Barbara Birk for technical assistance, Shuhui Zhao for help with platelet isolation, Michael Paolillo for reading the manuscript as well as all members of the Feil laboratory for constructive discussion. This work was supported by the Fund for Science, Deutsche Forschungsgemeinschaft (FOR 2060 projects FE 438/5-1 and FE 438/6-1, KFO 274 projects FE 438/7-1 and FE 438/8-2), and a travel grant from Boehringer Ingelheim Fonds.

## Author contributions
L.W. performed most experiments and participated in experimental design, data analysis, and manuscript writing. S.F., M.W., M.T., K.S. and C.W. helped with animal studies, in particular with the mouse cremaster thrombosis models and intravital FRET imaging. A.F. provided essential reagents. M.O., H.L. and M.G. helped with the flow chamber thrombosis model. F.R. performed western blot analyses. All authors provided constructive suggestions to experimental design and revised the manuscript. R.F. directed the study and wrote the majority of the manuscript.

## Additional information

**Competing interests:** The authors declare no competing interests.

