## [Peer Review File · Nature Communications]

Parts of this peer review file have been redacted as indicated to maintain the confidentiality of unpublished data.

Reviewers' Comments:

Reviewer #1:

Remarks to the Author:

By the use of a FRET-based probe for cGMP and transgenic mice thereof, Wen et al. show that flow/shear stress is required for NO-induced cGMP increase in platelets that form thrombus. With the flow chamber assay, the authors observed that cGMP in thrombus was increased only when the NO-containing buffer flows and decreased immediately after the stop of flow. To examine this phenomenon in physiological context, NO-dependent increase in cGMP was also observed at the mechanical or laser injury-induced thrombus in vivo. Interestingly, the increase in cGMP is observed primarily at the periphery of thrombus. With additional experiments with various conditions, it is proposed that flow/shear stress is required for activation of the NO-cGMP-cGKI cascade.

The work has been well done in an organized way and sheds a new light on the activation of NO-cGMP-cGKI cascade. It should also be highly evaluated that the ex vivo finding has been validated, albeit not entirely, by in vivo imaging with transgenic mice expressing the probe for cGMP. Major concerns are mostly on the interpretation of the results. The authors sometimes describe the "flow/shear stress" and sometimes just "flow". In the title, however, the authors proposed the NO-cGMP-cGKI cascade depends on "shear". In fact, with the knowledge disclosed in this work, it is not clear which, flow or shear stress, plays the major role on the activation of NO-cGMP-cGKI pathway.

Major comments:

1. Which of the flow or shear stress plays the major role in the activation of NO-cGMP-cGKI cascade? It is believed that NO diffuses very rapidly, but this concept is based on old in vitro observations. Thus, it is possible that NO was simply depleted around the platelet after the shutoff of flow. There are many NO probes. It may be useful to show that the NO concentration within platelets remains high after the shutoff of flow, if the authors persist to the shear-induced cGMP elevation.
2. A critical data missing in this paper is the cGMP level in the platelets floating in the DEA/NO solution. If the proposed model is correct, one would expect that DEA/NO will not increase cGMP in the platelets before adherence. The simplest experiment would be to measure CFP/YFP ratio of platelets before and after the addition of DEA/NO. There are reports using ELISA. To the best of this reviewer's knowledge, NO producer increases cGMP within platelets in vitro. Please comment on the difference between previous observations and present result about the effect of DEA/NO on platelets.
3. The data shown in in Fig. 3 is very interesting and important to integrate in vitro knowledge to physiology. However, the data may not support authors' proposal. Even before DEA/NO application the cGMP level is already high at the periphery of platelets (Fig. 3a-b). The increase in cGMP shown in Fig. 3c was correlated the increased CFP/YFP ratio within the platelets, where shear stress would be lower than the periphery. Thus, the result only says that the stimulant increases cGMP within the thrombus, but not more.
4. Very similarly, in Fig.3d-j, can the author detect the increase in cGMP immediately after the trapping of new platelets? The high cGMP at the periphery of thrombus may be observed because cGMP was already high before platelets adhere to the primary thrombus. How can the authors claim that cGMP in the platelets is elevated after the adherence to the injured arterioles?
5. Hiratsuka et al. recently reported PKA activation in the thrombus (J Thromb Haemost. 2017 Jul;15(7):1487-1499. doi: 10.1111/jth.13723). They also used transgenic mice expressing FRET probes. Considering the cross-talk between cAMP and cGMP pathways, this paper should be referred and discussed. Particularly, they proposed PKA is activated from the center of thrombus, to dissolve the aggregation. This observation contrasts to the authors' observation that cGMP is increased at the periphery of the platelets, although the time course shown in Fig. 3c, 3f, and 3i are similar to the increase in PKA activity. Furthermore, in comparison to the Hiratsuka's work, the present work lacks negative controls for the FRET probe. The inverse correlation of CFP and YFP

strongly argues for the authenticity of the FRET measurement, but may not be sufficient to show that the CFP/YFP ratio change detected in this study is free from some artifacts.

Minor comments:

1. It is recommended to include CFP/YFP image in movie 3, as in movie 2. It will give more vivid images of cGMP increase during thrombus formation.
2. Page 4, 3rd lines from the bottom; Fig. 3f should read as Fig. 2f.
3. Fig. 3f: It is said that the periphery of the region was measured for CFP/YFP. Please describe more precisely. Because the high CFP/YFP area is observed always, the line plot does not match to the impression of the video image, if the ratio reflects that of periphery of the thrombus.
4. In the figures of FRET images, the ratio ranges are currently shown just by "low" and "high". They should be shown numerically.

Additional notes:

1. The statics has been applied properly.
2. The methods are described in a way other researchers can reproduce.

Reviewer #2:

Remarks to the Author:

The authors have performed an interesting study that makes use of a genetically engineered reporter mouse to detect intracellular changes in platelet cGMP generation in real-time. They have found a novel shear-stress dependent amplification of NO-sGC induced cGMP formation, and identify this as a mechanism whereby increased levels of intravascular shear stress provide an NO-dependent brake on platelet aggregation and thrombus formation. Low flow would not engage this pathway.

The studies are elegant and well presented. However, there are a few missing pieces and seeming contradictions with existing literature, and these should be addressed.

1) As shown in the authors final scheme, platelet aggregation would normally occur on a bed of endothelial cells that themselves have a potent and well described shear-stress dependent NO-signaling system. This both responds via integrin and FAK mechanosensors via an Akt-eNOS phosphorylation pathway, as one primary mechanism. The NO generated would itself be anticipated to impact the platelets, and the question is the relative importance of the platelet-shear stress response to that of the underlying endothelial cells. The authors used a collagen substrate for their studies, and one wonders what would have happened if the underlying layer were an endothelial cell monolayer instead. Ideally, these cells would not have the FRET sensor.

2) The mechanism by which shear is supposed to activate sGC NO responsiveness is not examined. Is this via glycoprotein (GP)Ib-IX receptor complexes as has been suggested, or some other mechanism? Is there an eNOS involved, and is there like in endothelial cells, any role of a post-translational activation of eNOS. Does the mechanotransmission itself change as the shape of the platelet in the shear field is transformed, or as a function of the tensile stiffness of the platelet membrane? These would be anticipated to change as a function of the thrombus being formed.

3) Higher levels of shear have been linked to platelet activation (e.g. J Biomech. 2017 Jan 4;50:20-25., and a number of papers and reviews on the role of higher shear stress and thrombus formation - e.g. The impact of blood shear rate on arterial thrombus formation; Future Sci OA. 2015 Nov 1;1(4):FSO30. doi: 10.4155/fso.15.28. This seems opposite to the current conclusion that shear stress is enhancing NO-cGMP antithrombotic behavior. The authors need to examine this disparity, and relate their results to the large body of literature related to mechanical shear forces and platelet aggregation and thrombogenesis.

Reviewer #3:

Remarks to the Author:

This manuscript has two sides: On one hand, it reports an interesting platelet-specific cGMP reporter to detect cGMP in vitro and in vivo and demonstrate a rather exciting flow-dependent elevation in cGMP in response to NO in platelets. On the other hand, the part of study on the biological roles of cGMP in thrombosis is relatively poorly developed and preliminary, and the conclusions do not appear to explain all data in the field and in the manuscript. There are also some technical problems that may affect their results and conclusions. The authors are encouraged to make changes in the study, which may significantly enhance the novelty and significance of their study.

Major comments:

1. Although the controversy of the role of cGMP in platelets is to a degree associated with how to prepare platelets in vitro such as platelet pre-activation and desensitization etc, both the stimulatory role of relatively low concentrations of cGMP in platelet activation generated through endogenous platelet NO-dependent GC activation, and inhibitory role of high concentrations of cGMP induced by exogenous NO donors are both demonstrated by multiple groups of investigators in vitro and in vivo (including the JCI paper coauthored by Dr. Gawaz (coauthor of this paper), which should be cited). Fig 1 also suggest that inhibitory effect require cGMP induced by exogenous NO donors. Thus a better introduction in this difference would be beneficial to familiarize readers to the cGMP "controversy". Previous, Li et al (Cell 2013) proposed the "biphasic" theory of cGMP effect, and it appears that the finding of shear-dependent post-activation effect of NO/shear-cGMP may represent the second (inhibitory) phase of cGMP (also see comments below).
2. Heparin is associated with partial platelet activation and thrombocytopenia in mice. Thus, heparin as anticoagulant is problematic to study subtle regulatory mechanisms of platelets such as cGMP, which has opposing effect at different concentrations. More moderate increase in cGMP in platelets induced by agonists is likely to be submerged in the noise of baseline as a consequence.
3. In Fig 1 and 2, it is necessary to show cGMP levels with or without shear in the absence of exogenous NO as a control. Whereas it is interesting that flow elevates cGMP in response to exogenous NO, it is surprise that NO failed to induce cGMP at all in the absence of flow, which may suggest that there is a level of cGMP already before adding NO donor and flow.
4. Along the same line, it is a surprise that the flow dramatically reduce intracellular calcium level. In resting platelets, calcium level is at the bottom, hard to further decrease. These data indicate that the platelets are already activated when doing cGMP analysis. Thus, it is likely that cGMP levels and calcium levels have already been elevated before adding NO donors and flow. Also, platelet adherent to collagen is already activated. Hence the authors only analyzed additional changes in the cGMP and calcium levels in prior activated platelets. For this reason, the authors need to be careful not to over-interpret the implication of the data, and make it clear that their results do not address the role of cGMP during platelet activation.
5. It is known that shear stress, particularly the high shear stress, is important for platelet activation, elevation of calcium and aggregation during platelet adhesion. Here the authors showed that the shear stress is associated with decrease in calcium elevation, an indicator of platelet activation, which need better explanation and reconciliation with current knowledge. Combined with all above, it is likely that the observation by the authors is a post-platelet activation event.
6. In in vivo experiment, particularly in Fig 3f, elevation of cGMP precedes significant thrombus formation, and reduction in thrombus size is associated with decrease in cGMP level, making it

hard to argue that it is an inhibitory event. Also, the cGMP "level" is associated with increase in thrombus size, making it difficult to determine whether the FRET signals is due to cGMP elevation or simply the increase in number of platelets.

7. What is the sensitivity level of the cGMP report? Is it sensitive enough for more moderate increase in cGMP?

8. In "mechanic injury" experiment, it is not clear whether the pipette caused penetrating injury or not. It is not clear how to control the degree and size of damage. It is also not clear whether the accumulation of platelets is in a hole caused by the pipette on the vessel wall or in the lumen of the blood vessel. Interpretation of the results is different under these different conditions. Also, the movie should start before the platelet accumulation on the site of injury.

9. In figure 1, data do not reflect the effect of NO-cGMP-cGK1 on platelet aggregation, but is more likely to reflect stable platelet adhesion to collagen.

10. Is the flow-dependent VASP phosphorylation in Figure 1 cGK1-dependent?

11. The conclusion of NO/cGMP as a potent new anti-thrombotic is not fully supported by data, which only showed moderate effect during reversal phase in a relatively mild non-occlusive thrombosis model. The occlusive thrombosis, which is the target of anti-thrombotic therapy, usually do not have the reversal phase or causes damage before recanalization. Also, NO donors and cGMP elevating drug have been tried for many years as anti-thrombotic agents (without success) and therefore not new. With these being said, the finding of shear- and NO-dependent late inhibitory effect on thrombosis is in itself significant.

12. The moderate effect of high concentration of 8-br-cGMP (1 mM) on platelet adhesion or aggregation (which the assay does not differentiate) in the supplemental figure is previously known. The supplemental figure is not necessary for the paper, and should be removed.

13. Abstract and text needs to be more clear and easier to understand.

Reviewer #1 - expert in FRET biosensors (Remarks to the Author):

By the use of a FRET-based probe for cGMP and transgenic mice thereof, Wen et al. show that flow/shear stress is required for NO-induced cGMP increase in platelets that form thrombus. With the flow chamber assay, the authors observed that cGMP in thrombus was increased only when the NO-containing buffer flows and decreased immediately after the stop of flow. To examine this phenomenon in physiological context, NO-dependent increase in cGMP was also observed at the mechanical or laser injury-induced thrombus in vivo. Interestingly, the increase in cGMP is observed primarily at the periphery of thrombus. With additional experiments with various conditions, it is proposed that flow/shear stress is required for activation of the NO-cGMP-cGKI cascade.

The work has been well done in an organized way and sheds a new light on the activation of NO-cGMP-cGKI cascade. It should also be highly evaluated that the ex vivo finding has been validated, albeit not entirely, by in vivo imaging with transgenic mice expressing the probe for cGMP. Major concerns are mostly on the interpretation of the results. The authors sometimes describe the “flow/shear stress” and sometimes just “flow”. In the title, however, the authors proposed the NO-cGMP-cGKI cascade depends on “shear”. In fact, with the knowledge disclosed in this work, it is not clear which, flow or shear stress, plays the major role on the activation of NO-cGMP-cGKI pathway.

Major comments:

1. Which of the flow or shear stress plays the major role in the activation of NO-cGMP-cGKI cascade? It is believed that NO diffuses very rapidly, but this concept is based on old in vitro observations. Thus, it is possible that NO was simply depleted around the platelet after the shutoff of flow. There are many NO probes. It may be useful to show that the NO concentration within platelets remains high after the shutoff of flow, if the authors persist to the shear-induced cGMP elevation.

Response:

Thank you for pointing this out. Indeed, we have discussed this question for a long time. We talked to colleagues who have used NO probes such as DAF-FM. They did not recommend them, because these probes have limitations including irreversible reaction with NO. Therefore, under our flow on/off conditions, we would not detect a potential **decrease** of the platelet NO concentration during the experiment.

NO supply by release from DEA/NO is constant in the flow chamber throughout the flow on/off experiment (i.e. no matter whether flow is on or off). In the flow chamber, NO is freely diffusible. The NO concentration in/around a platelet might become lower after flow off only if this platelet consumes NO faster than new NO is being released from DEA/NO. Even if this would be the case, it is not evident to us why this NO “depletion” around platelets should be higher under flow off than under flow on condition.

We now added new data showing that cGMP is also flow/shear-sensitive in the presence of high DEA/NO concentrations (250 nM and 500 nM) (**page 4-5, line 177-181; new Supplementary Fig. 4**). At these high DEA/NO concentrations, it is unlikely that the cGMP drop upon flow off could be due to local NO depletion in/around the platelet. Another important finding is that we observed a similar flow/shear-sensitive cGMP regulation in the presence of the NO-GC stimulator Bay41-2272 (**Fig. 2d**). It is unlikely that this compound is “consumed” by platelets.

Altogether, our data strongly support the notion that shear stress rather than a potential flow-induced concentration change of NO is the factor that regulates cGMP generation in platelets. Hence, where appropriate we now use wording like “fluid shear stress”, “shear stress” or “shear” instead of “flow”.

2. A critical data missing in this paper is the cGMP level in the platelets floating in the DEA/NO solution. If the proposed model is correct, one would expect that DEA/NO will not increase cGMP in the platelets before adherence. The simplest experiment would be to measure CFP/YFP ratio of platelets before and after the addition of DEA/NO. There are reports using ELISA. To the best of this reviewer’s knowledge, NO producer increases cGMP within platelets in vitro. Please comment on the difference between previous observations and present result about the effect of DEA/NO on platelets.

Response:

Unfortunately, we are not able to perform FRET/cGMP imaging in floating platelets in suspension with our setup. However, we added new data of cGMP ELISA measurements of platelets in suspension under static or mild shaking condition (**page 5, line 187-196; new Supplementary**

Fig. 5c). In this experiment with “floating” platelets, we detected (a) that NO increased cGMP under static conditions and (b) that, although the shear generated during shaking is not well defined, the cGMP level was higher after shaking than under static conditions. However, the NO/shear-induced cGMP increase in floating platelets appeared to be weaker than in adherent platelets in a flow chamber. Our interpretation of these data is that there are clear differences in the shear-sensitivity of the cGMP-generating system in adherent (activated) and floating (presumably non-activated) platelets and that mechanosensitive cGMP generation is apparently more efficient in adherent than in floating platelets.

Note that the potential impact of shear stress has not been considered in previous cGMP measurements in platelets in suspension. However, these measurements generally used high concentrations of NO donors for stimulation and platelets were mixed during the procedure. Thus, it is likely that previous cGMP measurements in platelets were indeed performed in the presence of shear stress, although its magnitude was not known.

3. The data shown in Fig. 3 is very interesting and important to integrate in vitro knowledge to physiology. However, the data may not support authors’ proposal. Even before DEA/NO application the cGMP level is already high at the periphery of platelets (Fig. 3a-b). The increase in cGMP shown in Fig. 3c was correlated the increased CFP/YFP ratio within the platelets, where shear stress would be lower than the periphery. Thus, the result only says that the stimulant increases cGMP within the thrombus, but not more.

Response:

Fig. 3a-c shows a proof-of-principle cGMP measurement in an already stabilized thrombus that was then exposed to an NO donor (DEA/NO). The application of DEA/NO increased the cGMP level in the whole thrombus, i.e. periphery and core. It is likely that before application of DEA/NO, the stabilized thrombus was already exposed to endogenous NO supplied by the vessel wall (see **Fig 3d-i**). This might explain why in **Fig. 3b** the “baseline” cGMP level is higher in the periphery than the core. With the application of exogenous DEA/NO, the cGMP level increased further in the whole thrombus, both in the periphery and core.

Our conclusion that cGMP is higher in the periphery versus core is mainly based on **Fig. 3d-f** and **Fig. 3g-i**, where cGMP signals were observed in growing thrombi in the absence of exogenously added NO. These cGMP signals were stronger in the periphery of the thrombus than in its core region.

4. Very similarly, in Fig.3d-j, can the author detect the increase in cGMP immediately after the trapping of new platelets? The high cGMP at the periphery of thrombus may be observed because cGMP was already high before platelets adhere to the primary thrombus. How can the authors claim that cGMP in the platelets is elevated after the adherence to the injured arterioles?

Response:

As correctly mentioned by the reviewer, we cannot exclude that cGMP was already elevated before platelets adhered to the thrombus. With our intravital FRET imaging setup, we are not able to measure cGMP in floating platelets before they integrate into the thrombus. However, we consistently detected a higher cGMP level in the newly integrated platelets at the periphery than in the previously integrated platelets that were “relocated” to the thrombus core (**Fig. 3d-j**).

Importantly, the cGMP concentration in the periphery increased with the size of the thrombus. A bigger thrombus is exposed to higher shear stress. It is formally possible that the platelet cGMP concentration was already elevated before adhesion to the thrombus. However, in this case we would not expect a **dynamic increase** of the cGMP concentration during thrombus growth. Considering that (a) newly integrating platelets at the periphery are exposed to higher shear stress than those in the thrombus core, and (b) shear stress at the periphery dynamically increases during thrombus growth, it appears straightforward to conclude that the different cGMP levels in the periphery versus core were related to different levels of shear stress. In our manuscript we tried to make clear that cGMP in integrating platelets at the thrombus periphery is dynamically elevated during thrombus growth and that this correlates with the level of shear stress that they are exposed to (**page 7-8, line 305-328**).

5. Hiratsuka et al. recently reported PKA activation in the thrombus (J Thromb Haemost. 2017 Jul;15(7):1487-1499. doi: 10.1111/jth.13723). They also used transgenic mice expressing FRET probes. Considering the cross-talk between cAMP and cGMP pathways, this paper should be referred and discussed. Particularly, they proposed PKA is activated from the center of thrombus, to dissolve the aggregation. This observation contrasts to the authors' observation that cGMP is increased at the periphery of the platelets, although the time course shown in Fig. 3c, 3f, and 3i are similar to the increase in PKA activity. Furthermore, in comparison to the Hiratsuka's work, the present work lacks negative controls for the FRET probe. The inverse correlation of CFP and YFP strongly argues for the authenticity of the FRET measurement, but may not be sufficient to show that the CFP/YFP ratio change detected in this study is free from some artifacts.

Response:

Thank you for drawing our attention to the interesting paper of Hiratsuka and colleagues. We have cited (**new reference 36**) and discussed these findings now on **page 8-9, line 349-364**.

Hiratsuka et al. state on page 1493 of their paper that "In the propagation phase, PKA activity started to increase in platelets detaching from the downstream side of the aggregate, suggesting that high PKA activity is associated with thrombus resolution". This finding is indeed consistent with our data, indicating that high cGMP concentrations in shear-exposed platelets at the thrombus periphery limit thrombosis primarily through facilitation of thrombus dissolution. It is also interesting to note that PKA and cGMP-dependent protein kinase (named cGK in our manuscript) have overlapping substrate specificities. Thus, it is tempting to speculate that the substrate peptide in the PKA FRET sensor used by Hiratsuka et al. could also be phosphorylated by cGK.

The reviewer mentions the importance of controls to strengthen our intravital imaging data. We fully agree. A transgenic mouse line expressing a negative control cGMP sensor construct is not available and the generation of such a mouse line is beyond the scope of the present study. However, we have controlled our intravital FRET experiments in at least three ways to make sure we measured real cGMP signals. Firstly, as also acknowledged by the reviewer, FRET changes were confirmed by the inverse relationship of CFP and YFP traces (**Fig 3c,f,i**). Secondly, we used stimulation with NO donors as positive controls and observed robust cGMP signals in a spinning disk setup (**Fig 3a-c**) as well as by epifluorescence imaging (**Supplementary Fig.9**). A clear concentration-response relationship between FRET/cGMP signals and NO donor concentration was observed (**Supplementary Fig. 9**). Finally, and most importantly, we used NO-GC knockout mice as negative controls for intravital cGMP imaging. These mice are not able to generate cGMP in response to NO. Indeed, we did not observe FRET/cGMP changes in thrombi of these mice

(**Fig 3j,k**). Note also that we have performed similar negative controls with NO-GC knockout platelets for ex vivo FRET/cGMP imaging in the flow chamber (**Fig. 1f**).

Minor comments:

1. It is recommended to include CFP/YFP image in movie 3, as in movie 2. It will give more vivid images of cGMP increase during thrombus formation.

Response:

We have included the CFP/YFP ratio image as **new Supplementary Video 4**.

2. Page 4, 3rd lines from the bottom; Fig. 3f should read as Fig. 2f.

Response:

Thank you, done.

3. Fig. 3f: It is said that the periphery of the region was measured for CFP/YFP. Please describe more precisely. Because the high CFP/YFP area is observed always, the line plot does not match to the impression of the video image, if the ratio reflects that of periphery of the thrombus.

Response:

Please see also our responses to your major comments 3 and 4. With our FRET imaging setup we can monitor spatiotemporal changes between t1 and t2. Due to technical restrictions, we cannot monitor cGMP during initial thrombus formation. So we cannot say whether or not “baseline” cGMP at t1 was already elevated. It is likely that the thrombus at t1 was already exposed to endogenous NO and, thus, the platelets at the periphery would already generate cGMP (as can be seen in the video image at t1). However, further growth of the thrombus would expose the peripheral platelets to increasing shear stress, so that the cGMP concentration in the periphery further increases with thrombus growth. In the video images (**Fig. 3e**) it can be seen that the cGMP concentration in the periphery at t1 is indeed smaller than at t2, and this correlates with the line plot (**Fig. 3f**).

4. In the figures of FRET images, the ratio ranges are currently shown just by “low” and “high”. They should be shown numerically.

Response:

We now show the ratio ranges numerically in the FRET images of **Fig. 3b,e,h**.

Additional notes:

1. The statics has been applied properly.

2. The methods are described in a way other researchers can reproduce.

Reviewer #2 - expert in cGMP signalling (Remarks to the Author):

The authors have performed an interesting study that makes use of a genetically engineered reporter mouse to detect intracellular changes in platelet cGMP generation in real-time. They have found a novel shear-stress dependent amplification of NO-sGC induced cGMP formation, and identify this as a mechanism whereby increased levels of intravascular shear stress provide an NO-dependent brake on platelet aggregation and thrombus formation. Low flow would not engage this pathway.

The studies are elegant and well presented. However, there are a few missing pieces and seeming contradictions with existing literature, and these should be addressed.

1) As shown in the authors final scheme, platelet aggregation would normally occur on a bed of endothelial cells that themselves have a potent and well described shear-stress dependent NO-signaling system. This both responds via integrin and FAK mechanosensors via an Akt-eNOS phosphorylation pathway, as one primary mechanism. The NO generated would itself be anticipated to impact the platelets, and the question is the relative importance of the platelet-shear stress response to that of the underlying endothelial cells. The authors used a collagen substrate for their studies, and one wonders what would have happened if the underlying layer were an endothelial cell monolayer instead. Ideally, these cells would not have the FRET sensor.

Response:

We fully agree. According to the literature, endothelium-derived shear stress-dependent NO plays an important role in thrombosis. NO diffuses into nearby platelets and activates NO-GC to generate cGMP. However, our data indicate that endogenous NO (e.g. derived from the endothelium) would induce efficient cGMP production in platelets only if they are exposed to high shear stress (e.g. at the thrombus periphery). Therefore, shear stress is required at both levels, for NO synthesis in the endothelium and for NO-induced cGMP synthesis in the platelet thrombus. The experiment suggested by the reviewer (sensor-expressing platelets on sensor-negative endothelium) is already provided by our in vivo intravital FRET/cGMP measurements in growing thrombi (**Fig. 3 d-k**). We think under these close-to-native conditions, the endothelium-NO/platelet-cGMP pathway outlined above is indeed relevant. This has been stated in the text on **page 9, line 374-379**, and references on the role of shear-dependent NO release from the endothelium have been cited (**references 37-39**).

2) The mechanism by which shear is supposed to activate sGC NO responsiveness is not examined. Is this via glycoprotein (GP)Ib-IX receptor complexes as has been suggested, or some other mechanism? Is there an eNOS involved, and is there like in endothelial cells, any role of a post-translational activation of eNOS. Does the mechanotransmission itself change as the shape of the platelet in the shear field is transformed, or as a function of the tensile stiffness of the platelet membrane? These would be anticipated to change as a function of the thrombus being formed.

Response:

We are working hard to dissect the mechanism behind mechanosensitive cGMP signaling in platelets. As mentioned by the reviewer, it has been reported that vWF/GPIb-IX interaction leads to an increase of cGMP in platelets. Whether this process involves NO production in platelets is debated in the literature. In our FRET/cGMP measurements in the flow chamber, we did not detect

cGMP changes upon shear changes in the absence of exogenously added NO (**Fig.1e**, see beginning of the cGMP measurement, left part of the trace), suggesting that shear stress does not lead to NO generation in platelets, at least under our experimental conditions.

We appreciate the reviewer's questions and ideas about the mechanistic link between shear stress and cGMP production in platelets. However, these questions are particularly challenging to answer. We have now included new data showing that inhibition of several mechanosensitive ion channels or β_3 integrin does not affect the shear-regulated cGMP signals (**page 6, line 250-255; new Supplementary Fig. 7**).

[redacted]

3) Higher levels of shear have been linked to platelet activation (e.g. J Biomech. 2017 Jan 4;50:20-25., and a number of papers and reviews on the role of higher shear stress and thrombus formation - e.g. The impact of blood shear rate on arterial thrombus formation; Future Sci OA. 2015 Nov 1;1(4):FSO30. doi: 10.4155/fso.15.28. This seems opposite to the current conclusion that shear stress is enhancing NO-cGMP antithrombotic behavior. The authors need to examine this disparity, and relate their results to the large body of literature related to mechanical shear forces and platelet aggregation and thrombogenesis.

Response:

We acknowledge the vast body of literature showing that shear stress leads to platelet activation. Note that shear alone in the absence of NO did not affect cGMP and Ca²⁺ in our activated platelets. Thus, in order to limit thrombosis, shear **and** NO must be present. We think that shear-induced NO-cGMP signaling acts as an elegant endogenous “brake” (counterbalance, negative feedback) to prevent occlusive thrombosis without causing dangerous bleeding. We apologize if the manuscript was not clear in this regard and have revised the text accordingly (**page 9, line 374-385**).

Reviewer #3 - expert in platelet biology (Remarks to the Author):

This manuscript has two sides: On one hand, it reports an interesting platelet-specific cGMP reporter to detect cGMP in vitro and in vivo and demonstrate a rather exciting flow-dependent elevation in cGMP in response to NO in platelets. On the other hand, the part of study on the biological roles of cGMP in thrombosis is relatively poorly developed and preliminary, and the conclusions do not appear to explain all data in the field and in the manuscript. There are also some technical problems that may affect their results and conclusions. The authors are encouraged to make changes in the study, which may significantly enhance the novelty and significance of their study.

Major comments:

1. Although the controversy of the role of cGMP in platelets is to a degree associated with how to prepare platelets in vitro such as platelet pre-activation and desensitization etc, both the stimulatory role of relatively low concentrations of cGMP in platelet activation generated through endogenous platelet NO-dependent GC activation, and inhibitory role of high concentrations of cGMP induced by exogenous NO donors are both demonstrated by multiple groups of investigators in vitro and in vivo (including the JCI paper coauthored by Dr. Gawaz (coauthor of this paper), which should be cited). Fig 1 also suggest that inhibitory effect require cGMP induced by exogenous NO donors. Thus a better introduction in this difference would be beneficial to familiarize readers to the cGMP “controversy”. Previous, Li et al (Cell 2013) proposed the “biphasic” theory of cGMP effect, and it appears that the finding of shear-dependent post-activation effect of NO/shear-cGMP may represent the second (inhibitory) phase of cGMP\ (also see comments below).

Response:

The platelet cGMP controversy has been mentioned and referenced in the introduction of the original manuscript: “However, studies on the functional role of cGMP for platelet activity were inconsistent and, thus, the (patho-)physiological and therapeutic relevance of platelet cGMP signaling for hemostasis and thrombosis is debated¹⁴⁻¹⁷” (now **page 2, line 83-86**). We did not explicitly introduce the “biphasic” theory proposed by Li and colleagues, because our experiments did not address the initial phase of platelet activation that is supposed to be stimulated by cGMP. As correctly stated by the reviewer, we studied cGMP signaling in platelets that were already activated (reflected, for instance, by elevated Ca²⁺). Therefore, our experiments did not test the role of cGMP for platelet activation, but for later stages of thrombosis, which turned out to be inhibitory in our hands.

We have confirmed by P-selectin staining that our platelets were indeed activated (**page 4, line 133-135; new Supplementary Fig. 3c**).

As suggested by the reviewer, we have now introduced the concept of a biphasic role of cGMP and replaced the original references (letters to the editor) with recent review articles discussing these controversial issues (**page 2-3, line 86-92; new references 14-16**). We have also cited the JCI paper coauthored by Dr. Gawaz (**page 3, line 92, new reference 17**). We also state more clearly that we worked with activated platelets and that our data do not exclude a stimulatory role of cGMP during initial platelet activation (**page 2, line 58; page 8, line 346-347 and line 350-353**).

2. Heparin is associated with partial platelet activation and thrombocytopenia in mice. Thus, heparin as anticoagulant is problematic to study subtle regulatory mechanisms of platelets such as cGMP, which has opposing effect at different concentrations. More moderate increase in cGMP in platelets induced by agonists is likely to be submerged in the noise of baseline as a consequence.

Response:

We think the shear regulation of cGMP generation in platelets should not be affected by using heparin as anticoagulant. We also show a similar shear effect on Ca^{2+} in human platelets (**Supplementary Fig. 8c**), which have been isolated using a different anticoagulant (CPDA-1). We cannot exclude that we did not detect subtle increases in platelet cGMP that might be functionally relevant during platelet activation. Anyway, as stated above (response to reviewer comment 1), we did not look at initial platelet activation.

3. In Fig 1 and 2, it is necessary to show cGMP levels with or without shear in the absence of exogenous NO as a control. Whereas it is interesting that flow elevates cGMP in response to exogenous NO, it is surprise that NO failed to induce cGMP at all in the absence of flow, which may suggest that there is a level of cGMP already before adding NO donor and flow.

Response:

Indeed, the requested control experiment is very important and we have shown cGMP levels with or without shear in the absence of exogenous NO in **Fig.1e** (at the beginning of the cGMP measurement, left part of the trace). We stated in the manuscript on **page 4, line 167-168**: “Fourth, application of flow alone in the absence of NO did not evoke cGMP signals (**Fig. 1e, left**).”

We cannot formally exclude that the “baseline” cGMP level was already elevated in our activated platelets. However, we think this possibility is rather unlikely, because the NO-GC inhibitor ODQ did not reduce cGMP levels below “baseline” (**Fig. 2a-c**).

4. Along the same line, it is a surprise that the flow dramatically reduce intracellular calcium level. In resting platelets, calcium level is at the bottom, hard to further decrease. These data indicate that the platelets are already activated when doing cGMP analysis. Thus, it is likely that cGMP levels and calcium levels have already been elevated before adding NO donors and flow. Also, platelet adherent to collagen is already activated. Hence

the authors only analyzed additional changes in the cGMP and calcium levels in prior activated platelets. For this reason, the authors need to be careful not to over-interpret the implication of the data, and make it clear that their results do not address the role of cGMP during platelet activation.

Response:

We fully agree! We have now clearly stated that we analyzed cGMP and Ca²⁺ levels in prior activated platelets and that our results do not address the role of cGMP during initial platelet activation (**page 2, line 58; page 8, line 346-347 and line 350-353**).

5. It is known that shear stress, particularly the high shear stress, is important for platelet activation, elevation of calcium and aggregation during platelet adhesion. Here the authors showed that the shear stress is associated with decrease in calcium elevation, an indicator of platelet activation, which need better explanation and reconciliation with current knowledge. Combined with all above, it is likely that the observation by the authors is a post-platelet activation event.

Response:

We acknowledge the vast body of literature showing that shear stress leads to platelet activation. Note that shear alone in the absence of NO did not affect cGMP and Ca²⁺ in our activated platelets. Thus, in order to limit thrombosis, shear **and** NO must be present. We think that shear-induced NO/cGMP signaling acts as an elegant endogenous “brake” (counterbalance, negative feedback) to prevent occlusive thrombosis without causing dangerous bleeding. We apologize if the manuscript was not clear in this regard. We have revised the text accordingly (**page 9, line 374-385**). We also acknowledge that our study focusses on post-platelet activation events (**page 2, line 58; page 8, line 346-347 and line 350-353**).

6. In in vivo experiment, particularly in Fig 3f, elevation of cGMP precedes significant thrombus formation, and reduction in thrombus size is associated with decrease in cGMP level, making it hard to argue that it is an inhibitory event. Also, the cGMP “level” is associated with increase in thrombus size, making it difficult to determine whether the FRET signals is due to cGMP elevation or simply the increase in number of platelets.

Response:

It seems that cGMP and thrombus size increase in parallel. This is an interesting correlation, but as stated by the reviewer it is impossible to draw a conclusion on the functional relevance of cGMP from these data. This has also been stated in the original manuscript (**page 7-8, line 312-315**): “Interestingly, the magnitude of the cGMP signals correlated well with the area of the developing thrombus (**Fig. 3f**). These results indicated that an endogenous NO-cGMP system is active in platelet thrombi exposed to shear stress *in vivo*.” The functional relevance of platelet cGMP has been addressed in **Fig. 4**.

Regarding the question whether the FRET signal is indeed due to cGMP elevation or simply due to an increase in platelet number or other artifacts, the latter possibilities are highly unlikely. First, our FRET-based cGMP sensor measures the **concentration** of cGMP and not the “amount” of cGMP in a given region of interest. This means that a mere increase of the number of platelets (without a change of the intraplatelet cGMP concentration) would not result in a change of the FRET signal. Second, we have used an intramolecular ratiometric FRET sensor. Such sensors

are frequently used for quantitative measurements of signaling molecules, because their FRET signals are hardly influenced by changes in optical path length, excitation light intensity, biosensor expression level, or tissue movement/growth during data acquisition. Moreover, we have performed rigorous controls to make sure we measured indeed cGMP in our intravital FRET imaging experiments (please see also our response to reviewer #1, comment 5).

7. What is the sensitivity level of the cGMP report? Is it sensitive enough for more moderate increase in cGMP?

Response:

The cGi500 reporter has an EC_{50} for cGMP of ≈ 500 nM and a detection limit of ≈ 100 nM cGMP, which is well within the range of cGMP-binding affinities of known cGMP effector proteins, such as cGMP-dependent protein kinases (Campbell et al. 2017, ACS Chem Biol 12, 2388-98).

8. In “mechanic injury” experiment, it is not clear whether the pipette caused penetrating injury or not. It is not clear how to control the degree and size of damage. It is also not clear whether the accumulation of platelets is in a hole caused by the pipette on the vessel wall or in the lumen of the blood vessel. Interpretation of the results is different under these different conditions. Also, the movie should start before the platelet accumulation on the site of injury.

Response:

In mechanical injury-induced thrombosis experiments, a pipette with a diameter of ≈ 1 μm was used to puncture the vessel wall and induce the formation of a thrombus. The thrombus was growing in the vessel lumen (vessel walls are indicated with dashed lines in the figures and movies). Due to the mechanical injury and the resulting vessel movement, we could not achieve FRET imaging in the initial platelet adhesion stage. We captured thrombus growth starting from a medium-sized thrombus (**Fig. 3d-f, Supplementary Video 2**). For the technical reasons outlined above, the movie did not start before platelet accumulation.

With the mechanical injury model, it is indeed difficult to control degree and size of damage. Therefore, we switched to the laser-induced thrombosis model, which is technically more controllable and better suited for quantitative analysis of thrombus growth (**Fig. 4**). As documented in **Fig. 3g-j**, we obtained similar cGMP data with the laser-induced thrombosis model as with the mechanical injury model.

9. In figure 1, data do not reflect the effect of NO-cGMP-cGK1 on platelet aggregation, but is more likely to reflect stable platelet adhesion to collagen.

Response:

We have changed the title of **Fig. 1** and **Supplementary Fig. 1** accordingly.

10. Is the flow-dependent VASP phosphorylation in Figure 1 cGKI-dependent?

Response:

We did not test it. However, as we show that the flow-dependent Ca^{2+} change is mediated by cGMP/cGKI (**Supplementary Fig. 8a,b**), we anticipate that VASP phosphorylation would also be mediated by cGKI.

11. The conclusion of NO/cGMP as a potent new anti-thrombotic is not fully supported by data, which only showed moderate effect during reversal phase in a relatively mild non-occlusive thrombosis model. The occlusive thrombosis, which is the target of anti-thrombotic therapy, usually do not have the reversal phase or causes damage before recanalization. Also, NO donors and cGMP elevating drug have been tried for many years as anti-thrombotic agents (without success) and therefore not new. With these being said, the finding of shear- and NO-dependent late inhibitory effect on thrombosis is in itself significant.

Response:

Thank you for this assessment. We will keep this in mind.

12. The moderate effect of high concentration of 8-br-cGMP (1 mM) on platelet adhesion or aggregation (which the assay does not differentiate) in the supplemental figure is previously known. The supplemental figure is not necessary for the paper, and should be removed.

Response:

We have removed this supplementary figure and associated text (**page 7, line 274-278**).

13. Abstract and text needs to be more clear and easier to understand.

Response:

We have substantially revised the manuscript along the reviewer's comments.

Reviewers' Comments:

Reviewer #1:

Remarks to the Author:

Congratulations. The paper is greatly improved and the authors have answered to most of my concerns.

Reviewer #2:

Remarks to the Author:

The authors have addressed the concerns and improved the study. These results will be of considerable interest.

Reviewer #3:

Remarks to the Author:

This manuscript describes a novel and exciting finding: the NO-mediated platelet cGMP production is greatly facilitated by flow shear. The authors also narrow down their conclusion to the post-platelet activation phase and show that following the formation of a thrombus, the shear-dependent NO helps reducing the thrombus size. The revised manuscript is improved although there are some remaining problems that need to be addressed.

(1) The authors should make it even clearer in the abstract and conclusion that what they describe is a post-platelet activation NO production mechanism following the formation of a thrombus, as there are no data supporting that the same mechanism also applies to cGMP production during early platelet activation. Nor there is data suggesting that NO inhibited platelet activation in the early phase. In fact, the movie showing the newly added laser-induced thrombosis appear to demonstrate a faster and larger early thrombus formation in sGC knockout mice together with a faster reduction of the thrombus size in the later phase. It would be very helpful to provide a quantification and statistical analysis of that data. These data support "biphasic" effect rather than "inconsistent" previous results.

(2) The authors should provide sufficient details of quantification and statistical analysis, including sample size, normality test, and the method of statistical analysis in addition to p value.

(3) The statement of NO donors as a "potent" "new" anti-thrombotic drug should be either revised or supported by data, as NO donors are already clinically used and therefore not new, and there are no evidence suggesting that they are "potent" anti-thrombotic clinically. The authors also did not show the in vivo effect of NO donors in potently inhibiting thrombosis, using a more severe thrombosis model and even the mild model as used in the manuscript. The reviewer suggest to remove the words "potent" and "new", but instead state that this finding provide a potential new mechanism for the beneficial effects of NO donors in treating ischemic heart diseases.

Tübingen, 07.07.2018

Re: NCOMMS-17-26076A

Dear Reviewer #3,

Thank you for your additional comments on our revised manuscript.

Today, we have submitted our revised manuscript along with a point-by-point response to your new comments, see below (**original comments in bold**). We have also submitted a marked-up version of the revised manuscript highlighting changes to the previous manuscript in red. Page, line and figure numbers given in our response letter refer to the red-marked document.

Note that in order to comply with the format requirements for *Nature Communications*, we have shortened the abstract, separated the manuscript into Introduction/Results/Discussion sections, added subheadings in the Results section, and moved some other parts (Methods, References, etc.). Except for the abstract, these changes are also indicated in the red-marked manuscript.

Sincerely,

Robert Feil

Reviewer #3 (Remarks to the Author):

This manuscript describes a novel and exciting finding: the NO-mediated platelet cGMP production is greatly facilitated by flow shear. The authors also narrow down their conclusion to the post-platelet activation phase and show that following the formation of a thrombus, the shear-dependent NO help reducing the thrombus size. The revised manuscript is improved although there are some remaining problems that need to be addressed.

(1) The authors should make it even clearer in the abstract and conclusion that what they describe is a post-platelet activation NO production mechanism following the formation of a thrombus, as there are no data supporting that the same mechanism also applies to cGMP production during early platelet activation. Nor there is data suggesting that NO inhibited platelet activation in the early phase. In fact, the movie showing the newly added laser-induced thrombosis appear to demonstrate a faster and larger early thrombus formation in sGC knockout mice together with a faster reduction of the thrombus size in the later phase. It would be very helpful to provide a quantification and statistical analysis of that data. These data support “biphasic” effect rather than “inconsistent” previous results.

Response:

We have stated several times in both the abstract and the main text that we have studied NO-cGMP signaling in “activated platelets”. We now add a subheading in the Results section: “NO-cGMP signaling in activated platelets is shear-dependent” (**page 4, line 153**).

The reviewer states: **“In fact, the movie showing the newly added laser-induced thrombosis appear to demonstrate a faster and larger early thrombus formation in sGC knockout mice together with a faster reduction of the thrombus size in the later phase.”**

We cannot see this in sGC knockout mice. Is it possible that the reviewer swapped control and knockout mice? For further discussion of these data, please see below.

The reviewer states: **“It would be very helpful to provide a quantification and statistical analysis of that data.”**

Quantification and statistical analysis of data are shown in **Fig. 4b-g**. We have quantified data of $n = 31$ and $n = 37$ thrombi generated in 10 Ctrl and 7 NO-GC KO mice, respectively, as indicated in the figure legend. Statistical analysis is detailed in the Methods section (**page 15, line 640-648**). The tendency in **Fig. 4c** indicates a stimulatory role of NO-GC, but the tendency in **Fig. 4d** indicates an inhibitory role. Neither difference is statistically significant. The thrombus profile shown in **Fig. 4b**, averaging multiple thrombi/mice, does not show altered kinetics of initial thrombus formation in KO mice; the traces of control and KO are superimposed during the first ≈ 20 s. Instead, a significant difference between control and KO thrombi becomes evident at later stages (**Fig. 4b**). Further quantification shows that thrombus dissolution is significantly affected; the dissolution time is significantly longer in NO-GC KO than control mice (**Fig. 4e**). Moreover, the size of platelet thrombus integrated over time (AUC) as well as the size of stabilized thrombus are significantly bigger in the KO mice (**Fig. 4f,g**). Together, these data led us to the conclusion that, in our model, NO-cGMP facilitates thrombus dissolution in response to shear stress at the peak of thrombus formation.

The reviewer states: **“These data support “biphasic” effect rather than “inconsistent” previous results.”**

Our statement on “inconsistent results” in the Introduction referred to previous studies. We have deleted this statement (**page 2, line 79-80**).

(2) The authors should provide sufficient details of quantification and statistical analysis, including sample size, normality test, and the method of statistical analysis in addition to p value.

Response:

We have described the n-numbers etc. in each figure legend. Please note that the line traces show means \pm SEM. Details of statistical analysis are described in the Methods section (**page 15, line 640-648**). Evaluation of imaging data and thrombus growth is described in detail in the Methods section (**page 13-14, line 584-627**).

(3) The statement of NO donors as a “potent” “new” anti-thrombotic drug should be either revised or supported by data, as NO donors are already clinically used and therefore not new, and there are no evidence suggesting that they are “potent” anti-thrombotic clinically. The authors also did not show the in vivo effect of NO donors in potently inhibiting thrombosis, using a more severe thrombosis model and even the mild model as used in the manuscript. The reviewer suggest to remove the words “potent” and “new”, but instead state that this finding provide a potential new mechanism for the beneficial effects of NO donors in treating ischemic heart diseases.

Response:

We have removed the words “potent” and “novel” and added an additional sentence as suggested by the reviewer (**page 9, line 394 and line 396-399**).

Reviewers' Comments:

Reviewer #3:

Remarks to the Author:

Overall the paper is improved and interesting. Some minor modification would probably help further improvement.

In addressing the previous comment (1), the authors noted that they described several times that they have studied NO-cGMP signaling in activated platelets. The reviewer suggest that the authors use the term "pre-activated" platelets, instead of "activated" platelets, which may cover both immediate activation or post-activation. The authors in fact are studying "post-activation" cGMP regulation in platelets.

In previous comments, the reviewer referred to supplemental video 4, which showed a much faster and larger initial platelet thrombus in control as compared to sGC KO mice. Although the fluorescence in the video reflects cGMP levels, it appeared to also reflect the thrombus size and dynamics. Also please note a previous report suggesting "biphasic effect" of GC beta 1 ko (Zhang et al Blood 2011), although a different thrombosis model was used. Nevertheless, the reviewer noticed that the quantification of YFP fluorescence suggest no difference between control and KO using in during growing phase in the laser-injury model.

Tübingen, 07.09.2018

Re: NCOMMS-17-26076B

Dear Reviewer #3,

Thank you for your comments on our revised manuscript. Please find our response below.

Sincerely,

Robert Feil

Reviewer #3 (Remarks to the Author):

Overall the paper is improved and interesting. Some minor modification would probably help further improvement.

In addressing the previous comment (1), the authors noted that they described several times that that they have studied NO-cGMP signaling in activated platelets. The reviewer suggest that the authors use the term “pre-activated” platelets, instead of “activated” platelets, which may cover both immediate activation or post-activation. The authors in fact are studying “post-activation” cGMP regulation in platelets.

In previous comments, the reviewer referred to supplemental video 4, which showed a much faster and larger initial platelet thrombus in control as compared to sGC KO mice. Although the fluorescence in the video reflects cGMP levels, it appeared to also reflect the thrombus size and dynamics. Also please note a previous report suggesting “biphasic effect” of GC beta 1 ko (Zhang et al Blood 2011), although a different thrombosis model was used. Nevertheless, the reviewer noticed that the quantification of YFP fluorescence suggest no difference between control and KO using in during growing phase in the laser-injury model.

Response:

We followed the advice of the reviewer and now use the term “pre-activated” platelets instead of “activated” platelets throughout the manuscript.